# Drift in individual behavioral phenotype as a strategy for unpredictable worlds

Ryan T Maloney[1,2]*, Athena Q Ye[1], Sam-Keny Saint-Pre[1,3], Tom Alisch[1], David M Zimmerman[1,4], Nicole C Pittoors[1], Benjamin L de Bivort[1]

[1]Department of Organismic and Evolutionary Biology, Harvard University, Cambridge, United States; [2]Department of Psychology and Neuroscience, Colorado College, Colorado Springs, United States; [3]Tufts University, Medford, United States; [4]Department of Physics, Harvard University, Cambridge, United States

## eLife Assessment

Maloney et al. offer an **important** contribution to understanding the potential ecological mechanisms behind individual behavioral variation. By providing **compelling** theoretical and experimental data, the study bridges the gap between individual, apparently stochastic behavior with its evolutionary purpose and consequences. The work further provides a testable and generalizable model framework to explore behavioral drift in other behaviors.

**\*For correspondence:**
rtmaloney@coloradocollege.edu

**Competing interest:** The authors declare that no competing interests exist.

**Abstract** Individuals, even with matched genetics and environment, show substantial phenotypic variability. This variability may be part of a bet-hedging strategy, where populations express a range of phenotypes to ensure survival in unpredictable environments. In addition, phenotypic variability between individuals ('bet-hedging'), individuals also show variability in their phenotype across time, even absent external cues. There are few evolutionary theories that explain random shifts in phenotype across an animal's life, which we term drift in individual phenotype. We use individuality in locomotor handedness in *Drosophila melanogaster* to characterize both bet-hedging and drift. We use a continuous circling assay to show that handedness spontaneously changes over timescales ranging from seconds to the lifespan of a fly. We compare the amount of drift and bet-hedging across a number of different fly strains and show independent strain-specific differences in bet-hedging and drift. We show manipulation of serotonin changes the rate of drift, indicating a potential circuit substrate controlling drift. We then develop a theoretical framework for assessing the adaptive value of drift, demonstrating that drift may be adaptive for populations subject to selection pressures that fluctuate on timescales similar to the lifespan of an animal. We apply our model to real-world environmental signals and find patterns of fluctuations that favor random drift in behavioral phenotype, suggesting that drift may be adaptive under some real-world conditions. These results demonstrate that drift plays a role in driving variability in a population and may serve an adaptive role distinct from population-level bet-hedging.

## Introduction

No two organisms of the same species, even when genetically identical, behave precisely the same. Individuality has been observed and measured in organisms ranging from bacteria (*Beaumont et al., 2009*) to plants (*Cohen, 1966*), flies (*Buchanan et al., 2015*; *Linneweber et al., 2020*; *Honegger et al., 2020*; *Kain et al., 2015*; *de Bivort et al., 2022*) to humans (*Sanchez-Roige et al., 2018*), even in the absence of genetic or environmental differences. This variability poses two major questions in biology—how does it arise and what, if any, evolutionary role does it serve.

Heritable phenotypic variation in a population allows for adaptive tracking, that is when alleles change in frequency as the selective pressure of the environment changes (*Pfenninger and Foucault, 2022*). A complementary strategy for fluctuating environments is phenotypic plasticity, in which organisms change their behavior (or morphology) in direct response to environmental changes. These two adaptive strategies share a basic outcome: the phenotype they produce matches the current or recent environment. Organisms that fail to match the environment can suffer deadly consequences.

Random differences in phenotype may reflect a 'bet-hedging' strategy for species to deal with unpredictability in their environment (*Cohen, 1966*; *Hopper, 1999*; *Weissman et al., 2023*; *Ogura et al., 2017*; *Starrfelt and Kokko, 2012*; *Xue and Leibler, 2017*; *Müller et al., 2013*; *Sekajova et al., 2023*). Under this theory, variability allows some individuals to survive no matter what future environment arrives, increasing the odds that a population avoids extinction. Thus, bet-hedging species accept a lower arithmetic mean fitness for a higher geometric mean fitness (which equals zero if there is a single generation of no fitness). Theoretical work shows that bet-hedging strategies relying on random non-heritable variation outperform adaptive tracking in some environments (*Kain et al., 2015*; *Sekajova et al., 2023*; *Xue et al., 2019*). In particular, bet-hedging outperforms adaptive tracking when the environment fluctuates on timescales similar to the lifespan, as adaptive tracking requires multiple generations to respond to environmental cues (*Müller et al., 2013*; *Botero et al., 2015*), and can lag behind fluctuating selective pressures. While phenotypic plasticity occurs on faster timescales, the ability to adapt and learn can be metabolically costly and provides limited buffer against sudden changes in selective pressures (*Murren et al., 2015*). Intrinsic variability in genetic control (*Beaumont et al., 2009*) has been identified as a source of phenotypic variation in microbes. In multicellular organisms, stochastic processes in development underlie variation (*Honegger and de Bivort, 2018*), and, in the case of behavior, stochastic neuronal wiring (*Linneweber et al., 2020*; *Churgin et al., 2025*; *Mellert et al., 2016*; *Lillvis et al., 2022*) has been shown to predict persistent individual differences.

Individuals, however, do not express a singular behavioral phenotype across their life. Individual biases in multiple behavioral settings are only partially consistent over time (*Buchanan et al., 2015*; *Werkhoven et al., 2021*; *Honegger et al., 2020*; *Laskowski et al., 2022*; *Nakayama et al., 2016*; *Laskowski et al., 2022*; *Bierbach et al., 2017*), showing spontaneous changes even in the absence of macroscopic cues that could trigger plasticity. These observations raise two questions: to what degree does individuality arise due to developmental differences versus spontaneous fluctuations within the lifetime of the animal, and do fluctuations across the lifetime of an animal provide an evolutionary benefit? To answer these questions, we used near-isogenic animals to measure (1) the extent of variation present at the start of adulthood ('bet-hedging') and (2) the amount of behavioral phenotypic change over time ('behavioral drift'). We use locomotor handedness in *Drosophila melanogaster* as a model to measure how behavior drifts over time and show that the extent of behavioral drift is influenced by genes and the neuromodulator serotonin. We then use a life-history model to test the hypothesis that random change in phenotype ('phenotypic drift', as a more general form of behavioral drift) can be adaptive and assess whether real-world environmental fluctuations may drive the evolution of phenotypic drift. Taken together, these experiments and analyses suggest that phenotypic drift is plausibly an adaptive strategy to cope with environmental fluctuations within an organism's lifetime.

## Results

### Individual behavioral biases drift over time

To investigate the timescales on which individual preference spontaneously changes, we placed 252 flies in individual circular arenas and continuously tracked them for up to 30 days. At each time point, we computed the direction of circling (*Figure 1A*), a measure that exhibits idiosyncratic variation when averaged over long timescales (*Buchanan et al., 2015*). To look for slow changes in bias, we examined the turning bias of individual flies at different timescales: first, we averaged their direction of circling over each hour and second we low-pass filtered their continuous turning behavior to discard frequency components faster than 24 hr (*Figure 1B*). Comparing both of these measures to the experiment-wide average preference for each fly shows substantial long-duration changes from their experiment-wide circling tendency. To quantify this across all flies, we calculated the average power spectrum of the raw turning data across flies (*Figure 1C*), showing substantial power in lower

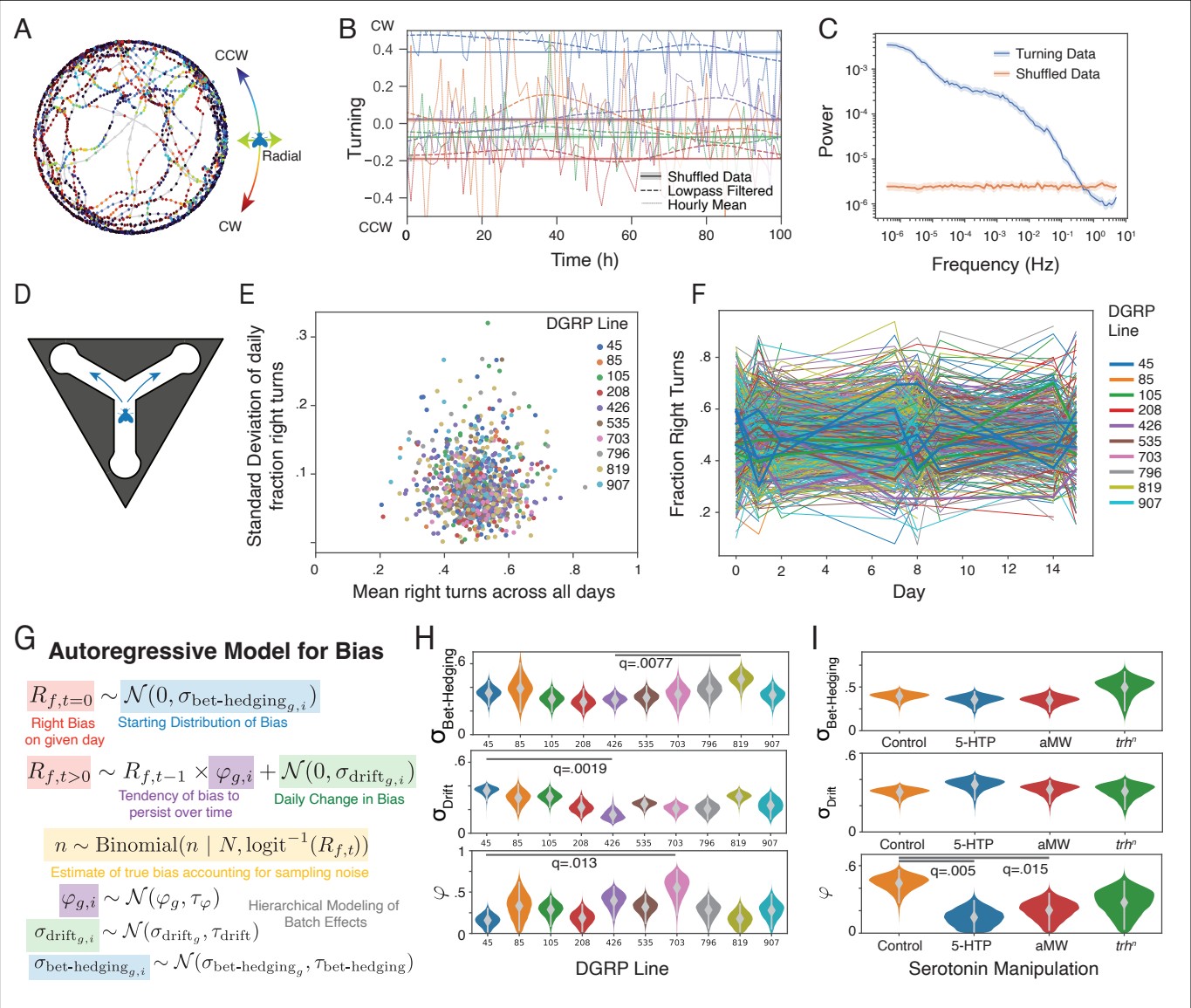

**Figure 1.** Characterizing changes in individual preferences in *Drosophila melanogaster*. (**A**) 2 hr sample of centroid-tracking data for a fly in a circular arena. Each point is colored based on whether it is moving CCW, CW, or radially in the arena. (**B**) Sample of 100 hr of continuous recording for 4 individual flies (colors). Hourly means of turning indices are shown in light lines. Dashed lines are low-pass filtered with a timescale cutoff of 24 hr. Solid lines are low-pass filtered shuffled data showing the average tendency across the experiment. (**C**) Mean power spectrum of turning data for all continuously monitored flies (n=252) for actual and shuffled data. Shaded areas represent 95% confidence intervals generated via bootstrapping (n=1000). (**D**) Schematic of Y-maze assay. Flies make either a left or right turn each time they walk through the intersection. (**E**) Standard deviation of daily right biases vs average right bias across days for individual flies (points). Colors indicate DGRP genotype (n=48–235 for each genotype (see *Table 1*)). (**F**) Mean individual handedness per day across all DGRP lines, a random subset are thickened to show representative changes in individual handedness over time. (**G**) Autoregressive model of individual right bias over time with parameters to estimate the initial right bias variability ($\sigma_{\text{Bet-Hedging}}$) and rate of daily change in right bias ($\sigma_{\text{Drift}}$). (**H**) Posterior estimates of $\sigma_{\text{Drift}}$, $\sigma_{\text{Bet-Hedging}}$, and $\phi$ (the autoregressive parameter characterizing the rate of reversion to zero bias), for each DGRP genotype. Grey bars represent 95% credible intervals. Lines indicate the smallest q value between populations. (**I**) As in H, posterior estimates of right bias variability parameters for flies treated with 5-HTP, AMW, and controls, as well as mutant flies with a missense mutation in *trh* generated by in vivo CRISPR. n=98 or 192 for each condition (see *Table 1*). q values <.05 are indicated.

The online version of this article includes the following figure supplement(s) for figure 1:

**Figure supplement 1.** Change in behavioral biases over time and the influence of genetics and serotonin on rate of change.

frequencies corresponding to hours- to days-long fluctuations in turning preferences. Interestingly, we did not see peaks at any specific frequencies (e.g. circadian), although there was a broad shoulder of power between $10^{-4}$ and $10^{-2}$ Hz. The overall trend across six orders of magnitude bore some resemblance to a power-law relationship between frequency and power. We saw similar patterns in other measures of behavior from this experiment, including speed, heading velocity, and distance from the center of the arena (*Figure 1—figure supplement 1A-D*).

## Genes regulate the rate of behavioral drift

The extent of behavioral variability in a population differs between genotypes (*Ayroles et al., 2015*; *Akhund-Zade et al., 2019*; *Akhund-Zade et al., 2020*). We next looked to see if the rate of behavioral drift also differs between genotypes. This is a prerequisite if behavioral drift can evolve as a trait under natural selection. We used 10 different lines from the *Drosophila* Genetic Resource Panel (DGRP) (*Mackay et al., 2012*). Using a Y-maze assay (*Figure 1D*) that measures left-right choices that correlate with locomotor handedness in circling (*Buchanan et al., 2015*), we measured the locomotor handedness of flies in each of these lines three times weekly for 3 weeks. As in the circling assay (*Figure 1A–C*), individual flies exhibited substantial variation in their daily average turn biases (consistent with drifting biases) in all genotypes, as measured by the standard deviation in the daily fraction of right turns for each fly, showing substantial variation from their experiment-wise average (*Figure 1E*), and evident in plotting the fraction right turns for each day of all flies (*Figure 1F*). We used a hierarchical Bayesian autoregressive model (*Figure 1G*) to estimate the initial variability in turn bias for each genotype ($\sigma_{\text{Bet-Hedging}}$), the extent to which turn bias changed each day ($\sigma_{\text{Drift}}$), and the tendency of turn bias to persist over time ($\varphi$; *Figure 1G*). We observed large differences in all three parameters between genotypes, as evidenced by minimally overlapping posterior distributions of estimates of these parameters (*Figure 1H*). All genotypes showed $\varphi$ above zero, demonstrating some level of persistence in individual biases over time, consistent with our circling experiments and previous studies of Y-maze handedness (*Buchanan et al., 2015*). Interestingly, we saw independent variation across genotypes in the posteriors for $\sigma_{\text{Bet-Hedging}}$ and $\sigma_{\text{Drift}}$, suggesting that different genetic mechanisms regulate behavioral bet-hedging and drift.

## Manipulating serotonin modulates the rate of behavioral drift

Neuromodulators have been shown to play an important role in regulating individuality in a large range of organisms (*Pantoja et al., 2020*; *Stern et al., 2017*; *Harel et al., 2024*; *Maloney, 2021*; *Sanchez-Roige et al., 2018*), including flies (*Kain et al., 2012*; *Honegger et al., 2020*; *Krams et al., 2021*). To test whether serotonin affects the rate of behavioral drift (and/or bet-hedging), we fed adult, isogenized Oregon-R flies food supplemented with either the serotonin synthesis inhibitor aMW or the serotonin precursor 5HTP and measured their locomotor handedness for 3 days each week for 3 weeks, as in the previous experiment. While $\sigma_{\text{Bet-Hedging}}$ and $\sigma_{\text{Drift}}$ were similar in all three treatment groups, both pharmacological manipulations of serotonin decreased the stability of behavior over time, increasing $\sigma_{\text{Drift}}$ and decreasing $\varphi$ (*Figure 1I*, *Figure 1—figure supplement 1F*). This was also evident in decreased correlation coefficients between handedness on successive days (*Figure 1—figure supplement 1H-J*) compared to control flies.

To assess the role of serotonin as a modulator of behavioral drift by a second approach, we generated a constitutive mutation in the tryptophan hydroxylase gene (*trh*), which is involved in the synthesis of serotonin. This approach also controls for off-target pharmacological effects and, unlike the previous experiment, deprives the animals of serotonin during development as well as after eclosion. We used in vivo CRISPR (*Zirin et al., 2020*) to knock out *trh* in an isogenized Oregon R background, yielding control flies of a closely matched genetic background. We did not see clear evidence of difference from control flies in the marginal distributions of $\sigma_{\text{Bet-Hedging}}$, $\sigma_{\text{Drift}}$, or $\varphi$ in our hierarchical model (*Figure 1I*, *Figure 1—figure supplement 1G*), although we did see a significant decrease of day-to-day correlation between control and *trh* null flies (*Figure 1—figure supplement 1K-M*).

## A theoretical adaptive advantage for phenotypic drift

The variation we see in $\sigma_{\text{Drift}}$ between genotypes suggests that behavioral drift can potentially evolve to provide fitness advantages in some situations. To build an intuition for the ideal phenotypic drift strategy and explore possible theoretical foundations, we formulated a simple two-state model

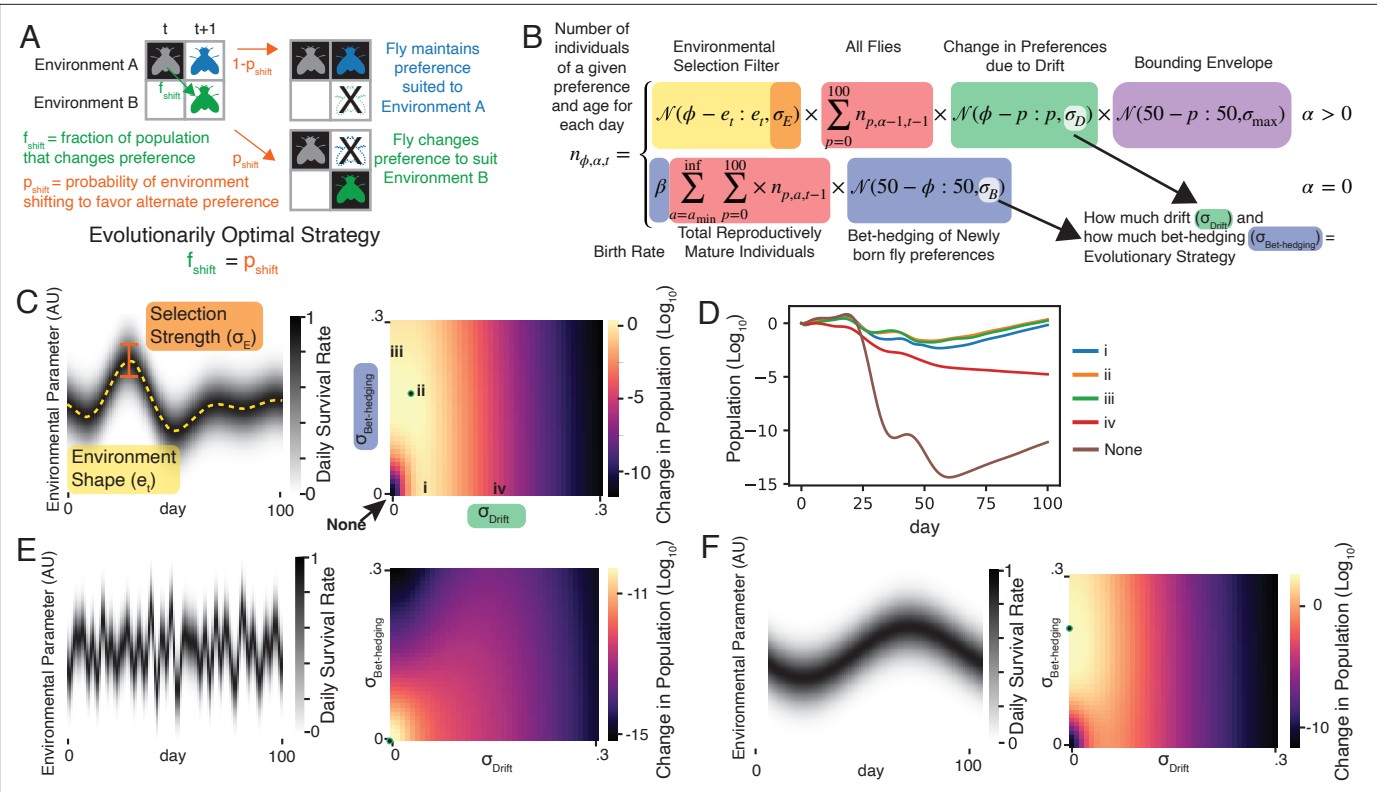

**Figure 2.** Modeling adaptive scenarios of phenotypic drift. (A) In a simplified model in which both phenotypes and environments have two states, the optimal fraction of the population that should change preference ($f_{\text{shift}}$) over a period of time equals the probability the environment changes ($p_{\text{shift}}$). See Appendix 1. (B) Model for the number of individuals with a particular continuous preference as a function of time in a fluctuating environment and individual age. $n_{\phi,\alpha,t}$, is the number of individuals with preference $\phi$, age $\alpha$ at day $t$ in the simulation. Two different cases determine this value, one for flies surviving from the previous timestep ($\alpha > 0$), and one for flies born in a particular timestep ($\alpha = 0$). The number of flies of a particular behavioral phenotype surviving on each successive day is determined by a function of how far that preference is from the ideal preference on that day (orange), the total number of flies that already have, or drift into having that phenotype on that day (red), and a bounding term (purple) that stops the distribution of preferences from diffusing away from the general range of what is adaptive (adding some degree of reversion to the mean consistent with our observed values of $\phi$, and via the Wiener-Khinchin theorem, consistent with the power spectrum observed in *Figure 1C*). The key behavioral strategy parameter from these terms is $\sigma_d$, which determines the rate at which flies' preferences drift over time. The number of new flies born each day is given by the total number of flies above the age of reproductive maturity $a_{\min}$ (red) times the birth rate $\beta$. New flies are born with an initial preference from a normal distribution centered on the long-term environmental mean with a standard deviation given by $\sigma_{\text{bet-hedging}}$ (blue) (C) Example environmental fluctuations and corresponding fitness landscape showing the change in population over 100 simulation days for differing amounts of phenotypic drift and bet-hedging. Green dot indicates ideal strategy (ii) (D) Population over time for strategies marked with Roman numerals in (C). (E, F) As in (C) for two additional example environmental fluctuation patterns.

The online version of this article includes the following figure supplement(s) for figure 2:

**Figure supplement 1.** Distribution of animal preferences over time for different strategies.

---

(*Figure 2A*). Imagine an organism in an environment that shifts probabilistically between two states. The organism can exhibit two behavioral phenotypes. If the chosen phenotype matches the environment, the organism survives; if not, it dies. What is the optimal strategy for changing its behavior to survive probabilistic shifts in the environment that occur with probability $p$? We can prove, through methods analogous to classic findings on optimal betting solutions (*Breiman, 1962*; *Kelly, 1956*) and previous analytical work on generational bet-hedging (*Tal and Tran, 2020*; *Bergstrom, 2014*), that in this two-state model, the optimal fraction of the population that should shift their behavioral phenotype in order to maximize long-term population growth equals $p$. This result extends to both the case of an arbitrary number of possible states as well as a continuous distribution of environmental states and behavioral phenotypes for all cases where the fitness narrowly depends on a tight match to a given environment (see Appendix 1 for proofs). Interestingly, this holds true even when the fitness associated with successfully matching phenotype to environment A differs from matching

environment B – in other words, matching the environment as much as possible is the most important thing, independent of the quality of specific environments.

## A flexible model of phenotypic drift shows a potential adaptive role under some patterns of environmental change

To model the effect of more realistic environmental fluctuations, we added dynamic environments, fitness effects, and life history to the auto-regressive model we used to fit the experimental behavioral data (*Figure 2B*). In this model, individuals, tracked at the population level, exhibit different behavioral preferences that can potentially vary over their lifespans. The environment fluctuates, and the survival probability of an individual increases as the difference between their preference and the environment decreases. Two parameters determine the phenotypic strategy: $\sigma_{\text{Bet-Hedging}}$, captures the initial variability of the population, and $\sigma_{\text{Drift}}$, captures the amount each fly's preference changed per day. By integrating this model over time, we estimated the fold-change in population size with varying levels of $\sigma_{\text{Bet-Hedging}}$ and $\sigma_{\text{Drift}}$ (*Figure 2C-D*, *Figure 2—figure supplement 1*), thereby determining the optimal variability strategy for a given pattern of environmental fluctuations. Sampling a few randomly generated environments quickly revealed that different patterns of fluctuation favored different strategies (*Figure 2E–F*), so we began to systematically assess what characteristics of the environment and life history might favor phenotypic drift, bet-hedging, or combinations of both.

## Phenotypic drift is an effective response at shorter timescale fluctuations than bet-hedging

To characterize the adaptive value of bet-hedging and phenotypic drift strategies as a function of statistics of environmental fluctuations, we created randomized environmental dynamics by filtering white noise in the time domain, leading to random fluctuations with a given temporal frequency content. We normalized and scaled the resulting environmental patterns and used them in simulations of population survival for a range of bet-hedging ($\sigma_{\text{Bet-Hedging}}$) and phenotypic drift ($\sigma_{\text{Drift}}$) values. From these simulations, we constructed a fitness landscape over the four dimensions of $\sigma_{\text{Bet-Hedging}}$, $\sigma_{\text{Drift}}$, environmental fluctuation period, and environmental fluctuation amplitude (*Figure 3A*). At very high-frequency fluctuations, the most successful strategy was $\sigma_{\text{Bet-Hedging}} = \sigma_{\text{Drift}} = 0$, reflecting an optimal strategy of all individuals tightly matching the average conditions of the environment (*Figure 3C*). Similarly, low amplitude fluctuations favored low bet-hedging and low drift. These findings make sense: low amplitude fluctuations are effectively a static environment that does not require a variable phenotypic strategy, and very rapid fluctuations changing within the timescale of organismic response are averaged away (*Kain et al., 2015*; *Müller et al., 2013*; *Botero et al., 2015*). However, as the amplitude of the fluctuations increased and the frequency decreased, the optimal strategy showed increased amounts of drift and bet-hedging, as producing individuals with a preference far from the environmental mean became increasingly necessary for survival. While both drift and bet-hedging are preferred over no variability in these situations, higher frequency fluctuations favor drift, while lower frequency fluctuations favor bet-hedging. Increasing the time to sexual maturity increases the advantage of drift, suggesting that varying a phenotype dynamically provides a mechanism for organisms to survive long enough to reproduce. These results suggest that the relative benefit of a drift or bet-hedging strategy depends on the rate of fluctuation of the environment compared to the development time of an organism. These relationships hold for a wide range of birthrates $\beta$ (*Figure 3—figure supplement 1A*) and ages of reproductive maturity $a_{\min}$ (*Figure 3—figure supplement 1B-C*).

## Real-world environmental fluctuations may favor phenotypic drift

To predict the adaptive value of phenotypic drift strategies in somewhat less synthetic circumstances, we collected 43 types of environmental time series from publicly available National Oceanography and Atmospheric Administration (NOAA; *Menne et al., 2012*) and National Ecological Observatory Network (*NEON, 2022*), National Ecological Observatory Network (NEON) datasets across 118,618 sites. We randomly sampled contiguous 1000-day time series from each of these datasets from the past 20 years, z-scored each time series, then scaled them to have standard deviations $\sigma_{\text{mean}}$ to represent different amplitudes of selective pressure. We then used these time series as model inputs to calculate the optimal amounts of phenotypic drift and bet-hedging as a function of $\sigma_{\text{mean}}$ and $a_{\min}$ (*Figure 4A–B*). Different patterns of environmental fluctuation, across differing measurement types

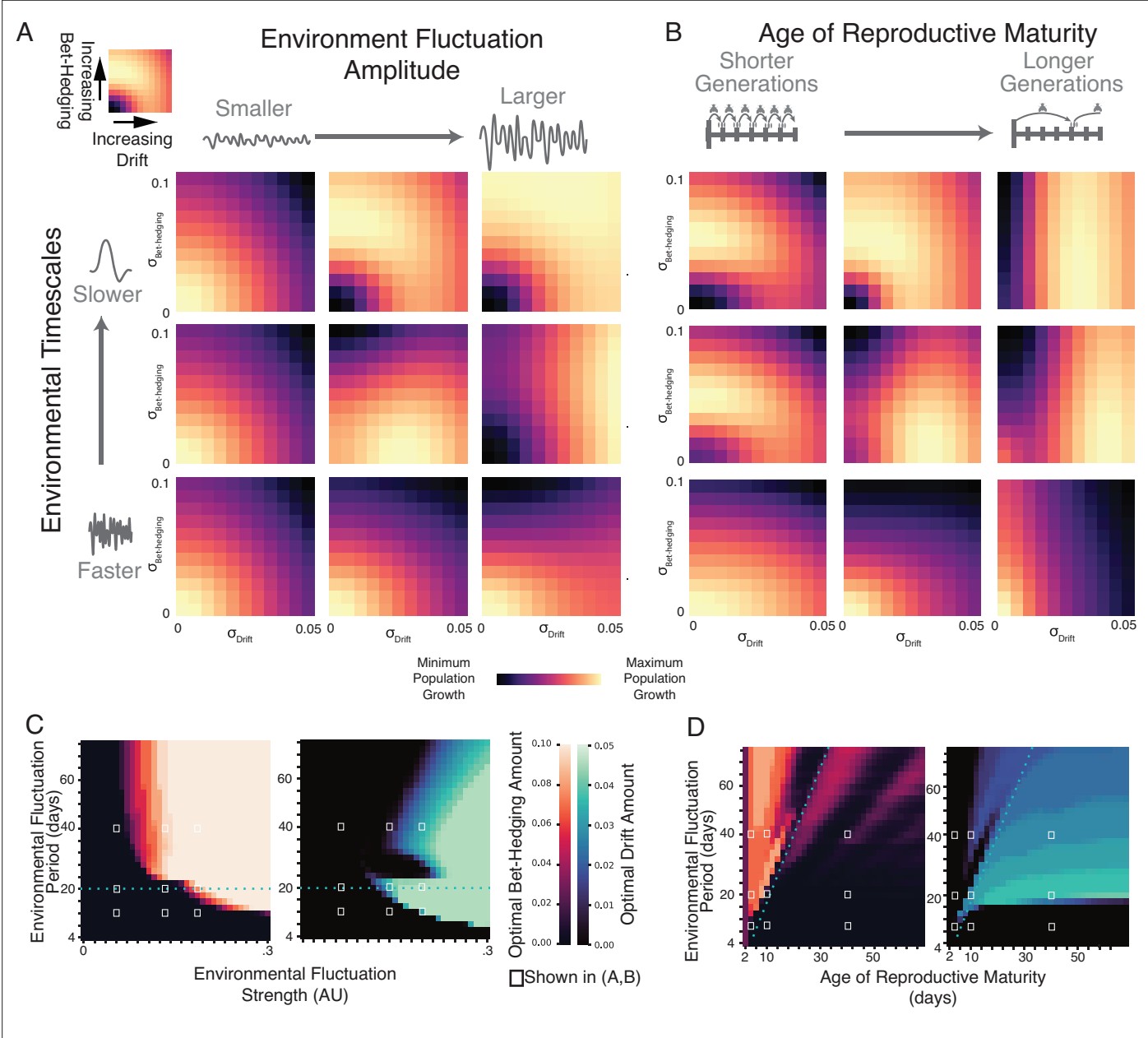

**Figure 3.** Effects of amplitude of environmental fluctuation, frequency of fluctuation, and age of reproductive maturity on ideal amounts of bet-hedging and phenotypic drift. (**A**) Fitness landscapes over combinations of environmental fluctuation amplitude and frequency. Each heatmap shows the geometric mean of the log change in population for each combination of drift and bet-hedging over 100 randomized environments. Heatmaps are normalized to their maximum and minimum values. Rows of heatmaps have the same environmental fluctuation frequency, and columns have the same environmental fluctuation amplitude (as measured by the standard deviation of all timepoints $\sigma_{Mean}$). The nine amplitude and frequency combinations in this panel correspond to values denoted with white boxes in (**C**). (**B**) As (**A**), except columns of heat maps have the same age of reproductive maturity, as determined by $a_{min}$. (**C**) Optimal amounts of bet-hedging (warm color scale; left) and drift (cool color scale; right) for each combination of environmental fluctuation amplitude and frequency. White squares indicate values associated with heatmaps in A. Dotted blue line corresponds to an environmental fluctuation period of 20 days, which is twice the age of reproductive maturity in these simulations. (**D**) As (**C**), except for optimal amounts of bet-hedging and drift for each combination of environmental fluctuation frequency and $a_{min}$. Dotted blue line corresponds to environmental fluctuations of twice the age of reproductive maturity. White squares indicate values associated with heatmaps in (**B**).

The online version of this article includes the following figure supplement(s) for figure 3:

**Figure supplement 1.** Effect of birthrate and longer timescales on strategies.

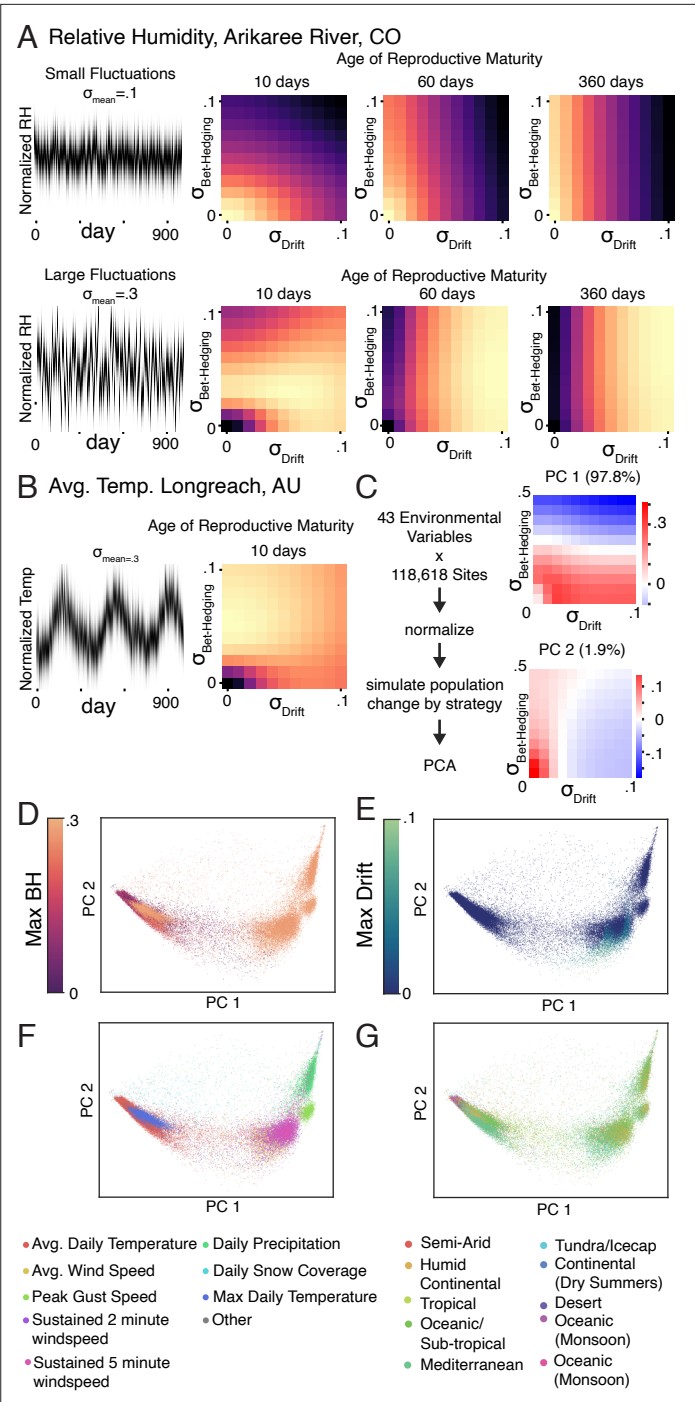

**Figure 4.** Optimal bet-hedging and phenotypic drift strategies for real-world environmental fluctuations. (**A**) 1000 days of relative humidity data from the Arikaree River in Colorado, USA (left panels) were used to generate an environmental selection filter with low (top-left) or high (bottom-left) amplitude fluctuations ($\sigma_{mean}$ in **Figure 2B**). Fitness landscape heatmaps over bet-hedging and drift strategies for different ages of reproductive maturity ($a_{min}$). (**B**) As in (**A**), using average daily temperature data from Longreach, Australia. (**C**) Pipeline for comparing the optimal variability strategies of organisms subject to real-world environmental fluctuations. Daily environmental time series from many sites were collected, normalized, and used in the model to produce fitness landscapes over $\sigma_{Bet\text{-}Hedging}$ and $\sigma_{Drift}$. All landscapes were then subject to principal components analysis. These simulations held $a_{min}$ and $\sigma_{mean}$ constant. See Methods. The loadings of PC1 (97.8% of the variance; top-right) indicate that this component encodes the optimal amount of bet-hedging, while PC2 (1.9% of the variance; bottom-right) encodes optimal drift. (**D**) Environmental time series from specific locations, plotted on PC2 vs PC1 axes, colored by

*Figure 4 continued on next page*

*Figure 4 continued*

optimal amount of bet-hedging. (**E**) As in (**D**), except color indicates optimal amount of drift. (**F**) As in (**D**), except color indicates the type of environmental measurement. (**G**) As in (**D**), except color indicates the Köppen climate classification of their location.

The online version of this article includes the following figure supplement(s) for figure 4:

**Figure supplement 1.** Breakdown of simulations by Koppen climate classification and measurement type.

and locales, showed differing optimal amounts of phenotypic drift and bet-hedging. As in our previous analyses, increasing $\sigma_{mean}$ increased the optimal $\sigma_{\text{Drift}}$ and $\sigma_{\text{Bet-Hedging}}$ (*Figure 3*), and increasing the $a_{\min}$ increased the optimal magnitude of phenotypic drift (*Figure 4A*, *Figure 4—figure supplement 1B,D*).

To relate optimal strategies of phenotypic drift and bet-hedging to climate and environmental factors, we used principal components analysis (PCA) to study variation in fitness landscapes over $\sigma_{\text{Bet-Hedging}}$ and $\sigma_{\text{Drift}}$ (*Figure 4C*). The first principal component of variability strategy landscapes had the vast majority of the variance (97.8%) and encoded the extent of bet-hedging; the second component (1.9%) encoded the extent of drift. Across time series, we observed a variety of optimal amounts of phenotypic drift and bet-hedging. A substantial minority (12%) of environmental signals favored non-zero drift (*Figure 4D and E*), particularly those reflecting temperature and wind-speed measurements and a subset of locations with tropical, sub-tropical, Mediterranean, and continental climates (*Figure 4F and G*). These results depend on the $a_{\min}$ in our model, with higher ages of reproductive maturity favoring more phenotypic drift (*Figure 4—figure supplement 1*).

## Discussion

We found that individual behavioral biases are not stable over an animal's lifetime, and that the degree of stability in behavioral bias is influenced by genetics and neuromodulation. Inspired by these findings, we provide a theoretical rationale for how instability in preferences could be adaptive for the long-term survival of individuals and species.

The behavioral preferences of individual flies change continuously over their life on days-long timescales (*Figure 1A–C*; *Figure 1—figure supplement 1A-D*). While indications of this phenomenon have been seen previously (*Buchanan et al., 2015*; *Honegger et al., 2020*; *Werkhoven et al., 2021*; *Yu et al., 2023*; *Bialek and Shaevitz, 2023*; *McKenzie-Smith et al., 2023*), these studies examined a small number of time points or recorded behavior continuously for up to a week. By measuring behavior continuously for up to 4 weeks (roughly the lifespan of a fly), we were able to estimate the complete power spectrum of behavior: the continuous distribution of timescales at which it changes. Across multiple behaviors (*Figure 1C*, *Figure 1—figure supplement 1A-D*), we show that there is no specific timescale over which behavioral preferences change (e.g. we did not see a circadian cadence to behavioral changes), and our data suggest that individuality shifts on a number of different time scales.

Examining several different inbred strains derived from a natural fly population, we found that the stability of preferences depends on genotype (*Figure 1H*). This implies that there is natural allelic variation that affects the rate of behavioral drift. In turn, this suggests that behavioral drift is not strongly maladaptive, in which case we would expect alleles involved in promoting behavioral drift to have been selected out of the population. Genetic variation for behavioral drift implies that the rate of drift could evolve to increase the fitness of animals subject to different kinds of environmental fluctuations. However, genetic variation for behavioral drift does not, per se, demonstrate that drift is subject to selection; drift could be a neutral epiphenomenon of some other trait under selection.

Importantly, some lines with differing magnitudes of initial preference variability ($\sigma_{\text{bet-hedging}}$) exhibited similar amounts of preference stability ($\sigma_{\text{drift}}$; e.g. lines DGRP 819 and 426). This suggests that mechanisms that drive differences in population preferences developmentally incompletely overlap with mechanisms that drive stability in preferences as adults.

Serotonin has been implicated as a regulator of the extent of individuality in fly behavior (*Kain et al., 2012*; *Honegger et al., 2020*; *Krams et al., 2021*) and stability of idiosyncratic preferences in *C. elegans* (*Stern et al., 2017*; *Ali Nasser et al., 2023*). Bidirectional pharmacological perturbation of serotonin signaling (increasing it with serotonin precursors, or decreasing it with synthesis inhibitors

or genetic perturbations) decreases the stability of preferences over time (or conversely increases the amount of behavioral drift; *Figure 1I*, *Figure 1—figure supplement 1F-M*). Our results from constitutive mutations in the serotonin synthesis pathway are less compelling (although we still see a significant difference in the strength of day-to-day correlations). The more modest impact of constitutive $trh^n$ mutation may reflect compensatory effects playing out across development. Together, our serotonin manipulation experiments suggest that this neuromodulator reinforces stability in idiosyncratic individual behavior. Further study is necessary to determine if serotonin regulates drift through its role in synaptic plasticity or some other widespread effect on the *Drosophila melanogaster* nervous system.

The combination of genetic and neuromodulatory control suggests that the extent of behavioral drift could evolve if it confers a fitness advantage in natural settings. The bet-hedging framework provides a theoretical basis for the adaptive value of stable idiosyncrasy in behavior (*Cohen, 1966*; *Simons, 2011*): namely, that variability in progeny phenotypes increases the chances that at least some offspring are fit when the environment fluctuates unpredictably. We hypothesize that behavioral drift may be advantageous for similar reasons, but operating within the lifespan of each individual. We provide two arguments in support of this hypothesis. First, we show in several analytically tractable cases (*Figure 2A*, Appendix 1) that it is evolutionarily optimal for an individual to switch behavioral preference phenotypes with a probability equal to the probability of the environment changing to favor those behavioral phenotypes.

This analytical result matches what we find using a more biologically realistic computational model of drifting individual preferences in fluctuating environments. We find that phenotypic drift is beneficial when the environment changes faster than the time to reproductive maturity of an animal. This finding suggests that behaviors that confer fitness with respect to aspects of the environment that change quickly should correspondingly change more quickly. Conversely, stable behavioral preferences may confer fitness with respect to stable aspects of the environment. This is borne out by simulating populations in different environments: quickly changing environmental parameters such as relative humidity (*Figure 4A*, *Figure 4—figure supplement 1D*) favor a drift strategy, while environments that vary at longer (e.g. seasonal) time scales, such as average daily temperature, favor a bet-hedging strategy (*Figure 4B*, *Figure 4—figure supplement 1D*). A key parameter of this model is the age of reproductive maturity ($a_{\min}$). Phenotypic drift provides a mechanism for organisms with slow development times to survive fluctuating environments long enough to reproduce. Thus, a key quantity is the time scale of environmental fluctuations relative to maturation time. We predict that organisms with more delayed onsets of reproduction will be more likely to exhibit phenotypic drift compared to organisms with more rapid development (subject to the same environmental fluctuations).

Organisms employ many strategies to survive changing environments. While our study focused on the respective advantages of bet-hedging (stable variability at the individual level) and phenotypic drift (variability within an individual's lifespan), these findings complement previous work comparing bet-hedging and adaptive tracking (selection-induced changes in allelic frequency over time *Machado et al., 2021*). Previous work on thermal preference in flies suggested that adaptive tracking offers an advantage when the environment fluctuates on approximately years-long time scales, whereas bet-hedging offers an advantage for months-long fluctuations (*Kain et al., 2015*; *Akhund-Zade et al., 2020*). These time scales roughly correspond to several fly lifespans and one lifespan, respectively. This study finds that phenotypic drift may be an adaptive strategy for environmental fluctuations within a lifespan. Thus, adaptive tracking, bet-hedging, and phenotypic drift appear to be complementary strategies for the challenges of environmental fluctuation over a wide range of time scales.

Importantly, random phenotypic drift and bet-hedging come at potentially high costs. If organisms could detect (or predict) environmental fluctuations and deterministically change their phenotype to be optimal for the realized environment, they could avoid the losses of randomly choosing the wrong phenotypes. Nonetheless, drift may play an adaptive role in some systems for two key reasons. Firstly, sensing and predicting future environments may be metabolically costly (*Murren et al., 2015*), especially for all possible environments. Random changes in preferences may therefore be more efficient than evolving the ability to reliably adapt to any fluctuation.

Second, random strategies may be inherently optimal. In a game-theoretic context, finite games always have an optimal strategy (a Nash equilibrium), but this strategy may have to be random. For instance, playing randomly in Rock Paper Scissors is optimal in so far as your opponent cannot learn

**Table 1.** *Drosophila melanogaster* genotypes used in this paper.

| Genotype | Source | Figure | n | Citation |
|---|---|---|---|---|
| Canton-S | BDSC 64349 | 1 A-C, S1A-D | 250 (24hr) 252 (2hr) | |
| DGRP 45 | BDSC 28128 | 1E, 1 G, S1E | 55 | *Mackay et al., 2012* |
| DGRP 85 | BDSC 28274 | 1E, 1 G, S1E | 22 | *Mackay et al., 2012* |
| DGRP 105 | BDSC 28139 | 1E, 1 G, S1E | 91 | *Mackay et al., 2012* |
| DGRP 208 | BDSC 25174 | 1E, 1 G, S1E | 111 | *Mackay et al., 2012* |
| DGRP 426 | BDSC 28196 | 1E, 1 G, S1E | 235 | *Mackay et al., 2012* |
| DGRP 535 | BDSC 28208 | 1E, 1 G, S1E | 115 | *Mackay et al., 2012* |
| DGRP 703 | BDSC 28218 | 1E, 1 G, S1E | 48 | *Mackay et al., 2012* |
| DGRP 796 | BDSC 28233 | 1E, 1 G, S1E | 140 | *Mackay et al., 2012* |
| DGRP 819 | BDSC 28242 | 1E, 1 G, S1E | 145 | *Mackay et al., 2012* |
| DGRP 907 | BDSC 28262 | 1E, 1 G, S1E | 141 | *Mackay et al., 2012* |
| Isod1 (Oregon R) | Clandinin Lab | 1H-I S(H-M) | 192 each (1 H: AMW, 5HTP, Control), 98 (1I: Control) | *Silies et al., 2013* |
| $trh^n$ | This study | 1I, S1 G,K-M | 98 | |
| GS01997 | BDSC 91886 | | | |
| Act-Cas9 | BDSC 54590 | | | |

to predict your play and exploit it. The randomness of stochastic evolutionary strategies, such as bet-hedging and phenotypic drift, may thus be truly optimal (*Tal and Tran, 2020*; *Bergstrom, 2014*). This may be particularly likely if the fitness effect of a particular behavioral phenotype depends on fluctuating game-theoretic interactions.

The interaction between behavioral drift and learning is likely complicated. Studies have observed idiosyncratic differences in learning that help shape the population-level distribution (*Smith et al., 2022*; *Manna et al., 2025*). Specifically, variability in response to identical cues may lead to variation in the population. Similarly, non-adaptive plasticity in response to unrelated cues has been shown to be a mechanism for generating differences in a population as part of a bet-hedging strategy (*Maxwell and Magwene, 2017*). While we limit our analysis in this paper to random shifts in phenotype rather than adaptive plasticity and learning, all these mechanisms of change likely co-exist in natural behaviors.

This paper characterizes changes in individual behavior within a fly's lifetime and proposes a theoretical framework in which such changes are evolutionarily adaptive. Together, these results motivate continued study of the biological mechanisms underpinning behavioral drift as well as empirical studies to test the hypothesis that behavioral drift helps organisms survive rapidly changing environments.

## Methods
### Fly care
Flies were grown on standard cornmeal/dextrose medium as previously described *Honegger et al., 2020*. Flies were kept on a 12:12 hr light:dark cycle at room temperature (20-23°). All behavioral experiments (excepting continual tracking experiments) were done during the animals' subjective day. Unless otherwise mentioned, all flies were 3–6 days post eclosion at the time of their first use

in a behavior experiment. Fly stocks used in this experiment are described in *Table 1*. To maintain individual identity across experiments, flies were either stored alone in standard media vials or in individual housing fly plates (modified 96 well plates *Alisch et al., 2018*; FlySorter, LLC).

## Continuous circling experiments

For continuous circling experiments, circular arenas were fabricated from three layers of laser-cut acrylic (floor, wall, and lid-holder layers) (*Werkhoven et al., 2021*). Lids were cut from 3-mm-thick clear acrylic. Wall layers were made from black acrylic and defined arenas of radius 28 mm and depth of either 1.6 mm or 10 mm. The latter arenas were filled with standard fly food until 1.6 mm of clearance remained. Flies were anesthetized by $CO_2$ and loaded singly into arenas, allowing 30 min of acclimation post-anesthetization. Transparent lids coated in sigmacote (Sigma-Aldrich) were placed on top of arenas to prevent fly escape and ceiling walking. Flies were recorded using a PointGrey Blackfly BFLY-PGE 12A2M camera at 30 frames per second for up to 14 days: either twice daily for 2 hr per session, or continuously for 24 hr. In the former experiment, fly identity was maintained between sessions by individual storage as described above. In the continuous experiment, flies were transferred to arenas with fresh food after anesthetization with $CO_2$. In 2 hr experiments, flies were given 30 min to recover before experiments. We used MARGO (*Werkhoven et al., 2019*) to record fly centroids in real time. All assays were conducted under 12:12 hr light and dark cycle conditions (9 AM:9 PM).

Power spectra were calculated using Lomb-Scargle periodograms (*VanderPlas and Ivezić, 2015*) to accommodate gaps in data due to lack of fly activity and periods when behavior was not recorded. Daily low-pass filtered data (*Figure 1B*) were created using a Blackman filter (*Blackman and Tukey, 1958*) with a window size of 51 hr, omitting missing data.

## Y-maze experiments

Y maze experiments were performed as described previously (*Buchanan et al., 2015*; *Werkhoven et al., 2019*). Briefly, flies were loaded individually into symmetrical y-maze arenas and their centroid locations were tracked using MARGO (*Werkhoven et al., 2019*), which detected each turn (moving from one arm of the y-maze to another) and calculated the fraction of right turns. All experiments were performed in white LED light in order to increase the activity of flies during the flies' subjective day photoperiod. Flies were anesthetized with $CO_2$ prior to loading into arenas; flies were allowed to acclimate for 20 min in the arenas prior to the beginning of the experiment. Handedness was computed based on the turns made in a 2-hr period each day; turns were scored based on the decision made each time the fly entered the center of the y-maze. Flies were cold-anesthetized for removal from the arenas, and identity was maintained as described above.

## Bayesian inference

Estimates for $\sigma_{bet\text{-}hedging}$, $\sigma_{drift}$, and $\varphi$ were generated using MCMC sampling in STAN 2.35 (*Carpenter et al., 2017*) according to the model in *Figure 1F*. Weakly informative priors were used as follows: $\sigma_{bet\text{-}hedging}$=Inverse Gamma($\alpha$=3, $\beta$=1); $\sigma_{drift}$=Inverse Gamma($\alpha$=3, $\beta$=1); $\varphi$=Normal($\mu$=0, $\sigma^2$=10) $R_{Missing}$=Normal($\mu$=0, $\sigma^2$=10). MCMC was performed with four chains, with the first 1000 draws discarded, posteriors were determined based on the subsequent 2000 draws. Inferred posteriors were robust to the choice of weakly informative priors.

Q values in 1 H,I were estimated based on the minimum posterior probability of the measured parameter having a difference of zero or less relative to the observed relationships based on the MCMC draws in STAN. In the case where the full simulated distribution had no overlap (*Figure 1H*, Phi), q values were given as q<1e-7.

## Pharmacology

For experiments where flies were treated with aMW, flies were collected as pupae and allowed to eclose on either treated or untreated food. Food was prepared and concentrations chosen based on previous reports (*Kain et al., 2012*; *Dierick and Greenspan, 2007*). For the aMW condition, aMW was added at a concentration of 20 mM with ascorbic acid used as a stabilizer (25 mg of ascorbic acid for every 100 mL of fly food). For the 5-HTP condition, 5-HTP was added at a concentration of

50 mM. Experiments (and corresponding control flies) were limited to flies who were at least 4 days post-eclosion at the start of the first timepoint. Food was replaced weekly.

## Generation of *trh^n* mutant

To generate a *trh^n* mutant in a defined genetic background, we used in vivo CRISPR (*Zirin et al., 2020*) to create *trh^n* mutants in the Isod1 background (an isogenized Oregon-R derivative; *Silies et al., 2013*). Briefly, Act-Cas9; Bl/CyO; TM2/TM6b flies were crossed to y/Y;U6-sgRNA flies (BDSC 91886) to produce +/y; +/CyO;*trh^n*/Tm6B flies. Mutations were verified via Sanger sequencing to generate a frame shift mutation at the targeted site, which were crossed back to +(Isod1);+(Isod1);Tm2/TM6b flies for multiple generations to produce +(Isod1);+(Isod1);*trh^n* flies, where all chromosomes except for the third chromosome (with the null mutant trh) were identical to IsoD1 controls. Putative mutant lines were genotyped via Sanger sequencing; *trh^n* lines used in this study were selected based on verification that non-homologous end joining induced a frame-shift mutation at the targeted point.

## Simulations

Simulations were implemented in Python 3 based on the model described in *Figure 2a*. All scripts are available at http://lab.debivort.org/drift-in-individual-preference/ and at https://zenodo.org/doi/10.5281/zenodo.13698148 and https://github.com/Maloney-Lab/Drift-in-Individual-Preference copy archived at *Maloney, 2026*. Changes in population were evaluated at steps corresponding to one day, and continuous distributions of preference were approximated with 200 preference bins ranging from an arbitrary scaling of -1 to 1. Variations in the amount of daily drift ($\sigma_{drift}$), dispersion of initial preferences ($\sigma_{drift}$), the bounding envelope ($\sigma_{max}$), the size of fluctuations in the environment ($\sigma_{mean}$), and the width of the environmental filter ($\sigma_{max}$) are all calculated on this [-1,1] range. Flies were assumed to have a maximum lifespan of 10 days plus twice the age of reproductive maturity for computational efficiency. Increasing maximum lifespan past this point did not quantitatively change the results of the simulations. Results for most simulations presented in the manuscript are available at the above URLs. For some results, only the environments are provided for space reasons, as the simulation is deterministic and can be rerun.

**Table 2.** Model parameters.

| Figure | $e_t$ (Environment series) | $e_t$ scaling time | $\sigma_e$ | $\sigma_D$ | $\sigma_B$ | $\sigma_{max}$ | $a_{min}$ | β (Birth Rate) | Sim. length (days) |
|---|---|---|---|---|---|---|---|---|---|
| *Figure 2C-2*, *Figure 3—figure supplement 1* | Temporally filtered white noise | 0.3 | 0.125 | 0–3 | 0–3 | 3 | 10 | 40 | 100 |
| *Figure 2E* | Temporally filtered white noise | 0.3 | 0.125 | 0–3 | 0–3 | 3 | 10 | 40 | 100 |
| *Figure 2F* | Temporally filtered white noise | 0.3 | 0.125 | 0–3 | 0–3 | 3 | 10 | 40 | 100 |
| *Figure 3A,C* | Temporally filtered white noise | 0–3 | 0.125 | 0–0.05 | 0–0.01 | 3 | 10 | 40 | 1001 |
| *Figure 3B,D* | Temporally filtered white noise | 0.3 | 0.125 | 0–0.05 | 0–0.01 | 3 | 2–72 | 40 | 1001 |
| *Figure 4A* | RH, Arikarree River (NEON) | 0.1, 0.3 | 0.125 | 0–0.1 | 0–0.1 | 3 | 10, 60, 360 | 40 | 1001 |
| *Figure 4B* | Avg. Temp, Longreach AU (NOAA) | 0.1, 0.3 | 0.125 | 0–0.1 | 0–0.1 | 3 | 10, 60, 360 | 40 | 1001 |
| *Figure 4C-G*, *Figure 3—figure supplement 1A* | NOAA/NEON | 0.2 | 0.125 | 0–0.1 | 0–0.5 | 3 | 10 | 40 | 1001 |
| *Figure 3—figure supplement 1B-C* | Temporally filtered white Noise | 0.3 | 0.125 | 0–0.1 | 0–0.1 | 3 | 2–1024 | 40 | 1001 |
| *Figure 4—figure supplement 1B,D* | NOAA/NEON | 0.1, 0.2, 0.3 | 0.125 | 0–3 | 0–3 | 3 | 10, 60, 360 | 40 | 1001 |

Environmental time-series ($E_t$) were created by generating random white noise time series and band-pass filtering them to the specified frequency bands in Python using the np.fft library. Resulting filtered time series were normalized to have a mean of zero and a standard deviation of $\sigma_{mean}$.

Detailed parameters of simulations used in figures are shown in *Table 2*.

## Real-world data collection

Real-world time series with 1-day (or better) resolution were collected from the Global Historical Climatology Network daily (GHCNd) and National Ecological Observatory Network (NEON) datasets. The GHCNd that is produced by the National Oceanic and Atmospheric Association (NOAA; *Menne et al., 2012*). Timeseries were collected from available stations across the world from 1998 to 2018. We also used data from the National Ecological Observatory Network (*NEON, 2022*), National Ecological Observatory Network (NEON). Data was accessed using the neonUtilities package (*Lunch et al., 2024*; see *Table 3*).

Time series used to run simulations were selected based on having at least 1000 days of continuous data with no gaps larger than 5 days: gaps were filled via linear interpolation. Time series with sampling more often than once per day were used to create average, maximum, and minimum daily value time series. Starting dates for sampled time series were chosen randomly among possible dates that met the above criteria such that no data point was used more than once. Due to the heterogeneous nature

**Table 3.** List of NEON datasets used.

| Product ID / DOI / data product name | Site IDs | Dates / date accessed |
|---|---|---|
| DP1.20217.001 https://doi.org/10.48443/br51-rd19 Temperature of groundwater | WLOU, WALK, TOOK, TOMB, SYCA, SUGG, REDB, PRPO, PRLA, PRIN, POSE, OKSR, MCDI, MAYF, MART, LIRO, LEWI, KING, HOPB, GUIL, FLNT, CRAM, COMO, CARI, BLWA, BLUE, BLDE, BIGC, BARC, ARIK | 2016-03-04 to 2022-04-30 DA: 06/28/2022 |
| DP1.00024.001 https://doi.org/10.48443/51ss-fm81 Photosynthetically active radiation (PAR) | WLOU, WALK, TOOK, TOMB, TECR, SYCA, SUGG, REDB, PRPO, PRLA, PRIN, POSE, OKSR, MCRA, MCDI, MAYF, MART, LIRO, LEWI, LECO, KING, HOPB, GUIL, FLNT, CUPE, CRAM, COMO, CARI, BLWA, BLUE, BLDE, BIGC, BARC, ARIK | 2016-03-04 to 2022-03-31 DA: 05/04/2022 |
| DP1.20261.001 https://doi.org/10.48443/jnwy-xy08 Photosynthetically active radiation below water surface | TOOK, TOMB, SUGG, PRPO, PRLA, LIRO, FLNT, CRAM, BLWA, BARC | 2017-07-28 to 2022-03-31 DA: 05/04/2022 |
| DP1.00004.001 https://doi.org/10.48443/rt4v-kz04 Barometric pressure | YELL, WREF, WOOD, WALK, UNDE, UKFS, TREE, TOOL, TOOK, TECR, TEAK, TALL, SYCA, SUGG, STER, STEI, SRER, SOAP, SJER, SERC, SCBI, RMNP, REDB, PUUM, PRPO, PRLA, PRIN, POSE, OSBS, ORNL, ONAQ, OKSR, OAES, NOGP, NIWO, MOAB, MLBS, MCRA, MCDI, MAYF, MART, LIRO, LEWI, LENO, LECO, LAJA, KONZ, KONA, KING, JORN, JERC, HOPB, HEAL, HARV, GUIL, GUAN, GRSM, FLNT, DSNY, DELA, DEJU, DCFS, CUPE, CRAM, CPER, COMO, CLBJ, CARI, BONA, BLUE, BLDE, BLAN, BIGC, BART, BARR, BARC, ARIK, ABBY | 2013-09-12 to 2022-01-31 DA: 03/13/2022 |
| DP1.00098.001 https://doi.org/10.48443/k9vk-5k27 Relative humidity | ABBY, ARIK, BARC, BARR, BART, BIGC, BLAN, BLDE, BLUE, BONA, CARI, CLBJ, COMO, CPER, CRAM, CUPE, DCFS, DEJU, DELA, DSNY, FLNT, GRSM, GUAN, GUIL, HARV, HEAL, HOPB, JERC, JORN, KING, KONA, KONZ, LAJA, LECO, LENO, LEWI, LIRO, MART, MAYF, MCDI, MCRA, MLBS, MOAB, NIWO, NOGP, OAES, OKSR, ONAQ, ORNL, OSBS, POSE, PRIN, PRLA, PRPO, PUUM, REDB, RMNP, SCBI, SERC, SJER, SOAP, SRER, STEI, STER, SUGG, SYCA, TALL, TEAK, TECR, TOOK, TOOL, TREE, UKFS, UNDE, WALK, WLOU, WOOD, WREF, YELL | 2013-09-12 to 2022-02-28 DA: 03/14/2022 |
| DP1.00023.001 https://doi.org/10.48443/9qpc-5v70 Shortwave and longwave radiation (net radiometer) | BART, UKFS, TALL, YELL, WREF, WOOD, WLOU, WALK, UNDE, TREE, TOOL, TOOK, TECR, TEAK, SYCA, SUGG, STEI, SRER, SOAP, SJER, SERC, SCBI, RMNP, REDB, PUUM, PRPO, PRLA, PRIN, POSE, OSBS, ORNL, ONAQ, OKSR, OAES, NOGP, NIWO, MOAB, MLBS, MCRA, MCDI, MAYF, MART, LIRO, LEWI, LENO, LECO, LAJA, KONZ, KONA, KING, JORN, JERC, HOPB, HEAL, HARV, GUIL, GUAN, GRSM, FLNT, DSNY, DELA, DEJU, DCFS, CUPE, CRAM, CPER, COMO, CLBJ, CARI, BONA, BLUE, BLDE, BLAN, BIGC, BARR, BARC, ARIK, ABBY | 2013-09-12 to 2022-05-31 DA: 06/07/2022 |
| DP1.00005.001 https://doi.org/10.48443/jqb2-vy96 IR biological temperature | YELL, WREF, WOOD, UNDE, UKFS, TREE, TOOL, TEAK, TALL, STER, STEI, SRER, SOAP, SJER, SERC, SCBI, RMNP, PUUM, OSBS, ORNL, ONAQ, OAES, NOGP, NIWO, MOAB, MLBS, LENO, LAJA, KONZ, KONA, JORN, JERC, HEAL, HARV, GUAN, GRSM, DSNY, DELA, DEJU, DCFS, CPER, CLBJ, BONA, BLAN, BART, BARR | 2013-09-12 to 2022-01-31 DA: 03/14/2022 |
| DP1.20288.001 https://doi.org/10.48443/t7rj-pk25 Water quality | TOOK, TOMB, SUGG, PRPO, PRLA, LIRO, FLNT, CRAM, BLWA, BARC, ARIK, MART, WALK, TECR, SYCA, REDB, PRIN, POSE, OKSR, MCRA, MCDI, MAYF, WLOU, LEWI, LECO, KING, HOPB, GUIL, CUPE, COMO, CARI, BLUE, BLDE, BIGC | 2014-01-11 to 2022-03-16 DA: 05/04/2022 |

of the observation sites with respect to time active, available sensors, and data quality, most data types were only available at a subset of sites.

Average Power Spectral Density in *Figure 4—figure supplement 1C* was calculated via Lomb-Scargle and averaged across all available time series.

## Acknowledgements

We thank M Miyagi, S Lavopulo, S Lall, D Lavrentovich, and other members of the de Bivort lab for helpful comments on this manuscript. This work was funded by NIH R01NS121874-01 to BdB and a Harvard Brain Institute Postdoctoral Pioneers grant to RM.

## Additional information

### Funding

| Funder | Grant reference number | Author |
|---|---|---|
| National Institute of Neurological Disorders and Stroke | R01NS121874-01 | Benjamin L de Bivort |
| Harvard Brain Institute | Postdoctoral Pioneers Award | Ryan T Maloney |

The funders had no role in study design, data collection and interpretation, or the decision to submit the work for publication.

### Author contributions

Ryan T Maloney, Conceptualization, Resources, Data curation, Software, Formal analysis, Supervision, Funding acquisition, Validation, Investigation, Visualization, Methodology, Writing – original draft, Project administration, Writing – review and editing; Athena Q Ye, Sam-Keny Saint-Pre, Tom Alisch, Investigation, Writing – original draft; David M Zimmerman, Formal analysis, Writing – original draft; Nicole C Pittoors, Investigation; Benjamin L de Bivort, Conceptualization, Resources, Supervision, Writing – original draft, Project administration, Writing – review and editing

### Author ORCIDs

Ryan T Maloney (ID) https://orcid.org/0000-0002-6111-7822
Tom Alisch (ID) https://orcid.org/0000-0001-7884-7031
David M Zimmerman (ID) https://orcid.org/0000-0002-8344-7072
Nicole C Pittoors (ID) https://orcid.org/0000-0001-6581-9568
Benjamin L de Bivort (ID) https://orcid.org/0000-0001-6165-7696

### Ethics

This study was exempt from review by the institutional IACUC committee as all work was done in the invertebrate Drosophila melanogaster.

Reviewer #1 (Public review): https://doi.org/10.7554/eLife.103585.3.sa1
Reviewer #2 (Public review): https://doi.org/10.7554/eLife.103585.3.sa2
Reviewer #3 (Public review): https://doi.org/10.7554/eLife.103585.3.sa3
Author response https://doi.org/10.7554/eLife.103585.3.sa4

## Additional files

### Supplementary files

MDAR checklist

## Data availability

All raw data, data acquisition software, and analysis scripts areavailableare available at http://lab. debivort.org/drift-in-individual-preference/ and https://zenodo.org/doi/10.5281/zenodo.13698148. Analysis scripts are available at https://github.com/Maloney-Lab/Drift-in-Individual-Preference (copy archived at *Maloney, 2026*). National Ecological Observatory Network (NEON) datasets used are listed in *Table 3*.

The following dataset was generated:

| Author(s) | Year | Dataset title | Dataset URL | Database and Identifier |
|---|---|---|---|---|
| Maloney R | 2024 | Data and Analysis Scripts for "Drift in Individual Behavioral Phenotype as a Strategy for Unpredictable Worlds" | https://doi.org/ 10.5281/zenodo. 13698148 | Zenodo, 10.5281/ zenodo.13698148 |

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

# Appendix 1

## Proof of optimal rate of switching phenotypes for two-state model

Consider a population facing the possibility of a shift in the environment with a probability $p_{shift}$. Each individual can choose to adopt a behavioral phenotype, a new behavioral phenotype, or stick with its current phenotype. If the phenotype that it chooses does not match the forthcoming environment, it will die. Otherwise, it will survive and reproduce with rate $\beta$. Evolution will maximize the long-term geometric growth rate.

### Proposition 1

In the case of two environments, the optimal fraction of progeny to shift to a phenotype adaptive to the different environment ($f_{shift}$) is equal to the probability of the environment changing to the other state ($p_{shift}$).

### Proof

The total change in population $G$, conditional on whether the environment shifts or stays, depends on the fraction $f$ of animals that shift or stay in their behavioral phenotype.

$$G_{shift} = \beta f_{shift},$$

$$G_{stay} = \beta f_{stay}.$$

Since population growth is multiplicative, the long-term change in population over $T$ generations, of which $T_{shift}$ have a change in the environment and $T_{stay}$ have no change is given by:

$$G_{total} = G_{shift}^{T_{shift}} G_{stay}^{T_{stay}},$$

and the long-term geometric average growth rate as

$$G_{avg} = (G_{shift}^{T_{shift}} G_{stay}^{T_{stay}})^{\frac{1}{T}}.$$

In the limit of large $T$,

$$T_{shift} = T p_{shift},$$

$$T_{stay} = T(1 - p_{shift}).$$

We can then write $G_{avg}$ in terms of $f_{shift}$ and $p_{shift}$:

$$G_{avg} = \beta f_{shift}^{p_{shift}} \cdot \beta (1 - f_{shift})^{1 - p_{shift}},$$

and the logarithm of average growth becomes

$$\log G_{avg} = 2 \log \beta + p_{shift} \log f_{shift} + (1 - p_{shift}) \log(1 - f_{shift}).$$

We can find the value of $f_{shift}$ where $G_{avg}$ is maximized by taking the partial derivative of $G_{avg}$ with respect to $f_{shift}$ to get

$$\frac{\partial \log(G_{avg})}{\partial f_{shift}} = \frac{(f_{shift} - p_{shift})}{f_{shift}(f_{shift} - 1)},$$

which equals zero when $f_{shift} = p_{shift}$. As $0 < f_{shift} < 1$, the second derivative is negative for all values of $f_{shift}$ and $p_{shift}$, thus $f_{shift} = p_{shift}$ maximizes $G_{shift}$.

## Proof of optimal distribution of phenotypes for discrete states

Consider the discrete-time stochastic process $\{e_t\}$ as a simple model of some fluctuating environmental signal. For the moment, we assume a finite set of possible environmental states. Further, we suppose

that the process is strictly stationary, so that all $e_t$ can be regarded as i.i.d. draws from the same categorical distribution ($p_e$) over environmental states $e \in \{1, 2, \ldots, m\}$.

Suppose that an organism expresses exactly one phenotype from birth, according to the categorical distribution ($f_\phi$) over phenotypes $\phi \in \{1, 2, \ldots, n\}$. Let $\beta_{e_t, \phi}$ be the fitness of phenotype $\phi$ under environmental state $e$, that is the subpopulation expressing phenotype $\phi$ will grow by a factor of $\beta_{e_t, \phi}$ during generation $t$. After $k$ generations (and assuming that the population size is so large as to be infinitely divisible), the total population will have changed by a factor of

$$G_k = \prod_{t=1}^{k} \sum_{\phi=1}^{n} \beta_{e_t, \phi} f_\phi, \qquad e_t \sim \text{Categorical}(p_1, \ldots, p_m). \tag{1}$$

## Diagonal fitness

Suppose that the fitness matrix $\beta = [\beta_{e\phi}]$ is diagonal ($\beta_{e,\phi} = \delta_{e,\phi}\beta_\phi$), and, for simplicity, square ($m = n$), so that only phenotype $\phi = e$ has nonzero fitness in a given environment $e$. We then have

$$G_k = \prod_{t=1}^{k} \beta_{e_t} f_{e_t}, \qquad e_t \sim \text{Categorical}(p_1, \ldots, p_n). \tag{2}$$

*Kelly, 1956* made the following crucial observation:

### Proposition 2 (Kelly)

The phenotype distribution ($f_e$) that maximizes the expected population fold change $\langle G_1 \rangle$ over a single generation is simply $f_e = \delta_{e\hat{e}}$, where $\hat{e} = \underset{e}{\text{argmax}}\, p_e$.

In other words, the strategy that maximizes the expected population increase over any finite time horizon is for *all* individuals to express the unique phenotype adapted to the maximally likely environment. This strategy will result in certain extinction as soon as a different environment occurs, which will happen almost surely in the limit that $k \to \infty$. For this reason, Kelly introduced an alternative measure of population fitness: namely, the time-averaged exponent of growth *Kelly, 1956*.

### Definition 1

We define the *asymptotic growth rate* to be

$$\lambda = \lim_{k \to \infty} \frac{1}{k} \log G_k, \tag{3}$$

and we say that any phenotype distribution ($f_e$) that maximizes $\lambda$ is *Kelly-optimal*.

The following theorem due to *Breiman, 1961* justifies the use of the Kelly criterion.

### Theorem 1 (Breiman)

A Kelly-optimal phenotype distribution is superior to all other betting strategies in two senses:

1. The growth rate of a Kelly-optimal strategy asymptotically dominates that of any other strategy almost surely.
2. A Kelly-optimal strategy minimizes the expected time for the population size to increase by an arbitrarily large amount.

### Proposition 3

The Kelly-optimal phenotype distribution for the case of a diagonal fitness matrix is $f_e = p_e$, independent of $\beta$.

### Proof

Using the law of large numbers, we can replace the time average in (3) by an expected value:

$$\lambda = \lim_{k \to \infty} \frac{1}{k} \log G_k \tag{4}$$

$$= \lim_{k \to \infty} \frac{1}{k} \sum_{t=1}^{k} \log(\beta_{e_t} f_{e_t}) \tag{5}$$

$$= \sum_{e=1}^{n} p_e \log(\beta_e f_e). \tag{6}$$

Which can be rearranged as follows:

$$\lambda = \sum_{e=1}^{n} p_e \log \beta_e + \sum_{e=1}^{n} p_e \log f_e \frac{p_e}{p_e} \tag{7}$$

$$= \underbrace{\sum_{e=1}^{n} p_e \log \beta_e}_{\langle \beta_p \rangle} + \underbrace{\sum_{e=1}^{n} p_e \log p_e}_{-H(p)} - \underbrace{\sum_{e=1}^{n} p_e \log \frac{p_e}{f_e}}_{D_{\mathrm{KL}}(p\|f)}. \tag{8}$$

These three terms are the expected value of $\beta_p$, the negative entropy of $p$, and the Kullback–Leibler divergence of $p$ from $f$, respectively. Only the third term depends on the phenotype distribution $f$. Since $D_{\mathrm{KL}}(p \parallel f) \geq 0$, with equality if and only if $f_e = p_e$, $\lambda$ is maximized when $f_e = p_e$.

The expression for $\lambda$ given in (8) has a very intuitive interpretation. To see this, we note that the first term is simply the asymptotic growth rate for a population whose members *all* perfectly match the environment at every generation. However, a population will in general do worse than this upper bound for two reasons: (1) the intrinsic unpredictability of the environment (reflected by a penalty of magnitude $H(p)$, the entropy of the environment), and (2) incomplete knowledge of the true environmental distribution (reflected by a penalty of magnitude $D_{\mathrm{KL}}(p \parallel f)$, the K–L divergence between the phenotype and environment distributions).

Yet another way of understanding the intuition behind Proposition 3 is motivated by considering the *worst-case* value of $\lambda$ under the Kelly-optimal phenotype distribution: that is by minimizing $\lambda_{\mathrm{Kelly}} = \langle \beta \rangle_p - H(p)$ w.r.t. the environment distribution ($p_e$). Using a Lagrange multiplier to enforce the normalization of $p$, one obtains

$$p_e^* = \underset{\|p\|=1}{\arg\min} \, \lambda_{\mathrm{Kelly}} = \frac{1}{z\beta_e}, \tag{9}$$

where $z = \sum_e \beta_e^{-1}$ is a normalizing constant. Evidently, the asymptotic growth rate is minimized when the probability of a given environment is inversely proportional to the fitness of the matching phenotype, so that the organism thrives only in low-probability environments. Using (9), we may then rewrite (8) in the form

$$\lambda = -\log z + \underbrace{\sum_{e=1}^{n} p_e \log p_e z \beta_e}_{D_{\mathrm{KL}}(p\|p^*)} - \underbrace{\sum_{e=1}^{n} p_e \log \frac{p_e}{f_e}}_{D_{\mathrm{KL}}(p\|f)}. \tag{10}$$

Thus, the fitness achieved by an organism reflects two opposing K-L divergences corresponding to how far the environmental distribution is from one that minimizes the Kelly-optimal growth rate ($D_{\mathrm{KL}}(p \parallel p^*)$) and how far the environmental distribution is from the organisms' phenotypic distribution ($D_{\mathrm{KL}}(p \parallel f)$). By analogy, fitness depends on the respective degrees to which an organism and its 'antagonists' (i.e. the forces that conspire to determine the fitness landscape $\beta$) are maximally informed about the distribution of environmental states. In the case that $z = 1$, the overall growth rate is positive exactly when the organism 'knows' more about the environment than the antagonists, in the sense that the environment distribution more closely matches the organism's phenotype distribution than it does the inverse fitness landscape. Moreover, we may regard the term $-\log z$ as a measure of 'excess fitness', or the degree to which the environment overall is biased in favor of the organism: if it is negative ($z > 1$), then the organism has to do *better* than the antagonists just to break even (i.e. to achieve $\lambda = 0$), while if it is positive ($z < 1$), then the organism can still achieve net positive growth even under the worst-case scenario that $p_e \propto \beta_e^{-1}$.

## Constant baseline fitness

Suppose that $\beta_{e,\phi} = \epsilon + \delta_{e,\phi}(\tilde{\beta}_\phi - \epsilon)$, corresponding to the case of a fitness matrix of the form

$$[\beta_{e,\phi}] = \begin{pmatrix} \tilde{\beta}_1 & \epsilon & \epsilon & \cdots & \epsilon \\ \epsilon & \tilde{\beta}_2 & \epsilon & \cdots & \epsilon \\ \epsilon & \epsilon & \tilde{\beta}_3 & \cdots & \epsilon \\ \vdots & \vdots & \vdots & \ddots & \vdots \\ \epsilon & \epsilon & \epsilon & \cdots & \tilde{\beta}_n \end{pmatrix},$$

where $\epsilon > 0$ is the (small) baseline fitness in the case of a phenotype–environment mismatch. The asymptotic growth rate is then given by

$$\lambda = \lim_{k \to \infty} \frac{1}{k} \log G_k \tag{11}$$

$$= \lim_{k \to \infty} \frac{1}{k} \sum_{t=1}^{k} \log \left( \sum_{\phi=1}^{n} \beta_{e_t,\phi} f_\phi \right) \tag{12}$$

$$= \sum_{e=1}^{n} p_e \log \left( \sum_{\phi=1}^{n} \beta_{e,\phi} f_\phi \right) \tag{13}$$

$$= \sum_{e=1}^{n} p_e \log \left( \tilde{\beta}_e f_e + \epsilon(1 - f_e) \right). \tag{14}$$

## Claim 1

To leading order in $\epsilon$, the Kelly-optimal phenotype distribution in the case of constant baseline fitness is given by

$$f_e \simeq p_e(1 + \epsilon z) - \frac{\epsilon}{\tilde{\beta}_e},$$

where $z = \sum_e \tilde{\beta}_e^{-1}$ as above.

## Proof

We seek a first-order solution to the following constrained optimization problem:

$$\min_{f} \quad \sum_{e=1}^{n} p_e \log \left( \tilde{\beta}_e f_e + \epsilon(1 - f_e) \right) \tag{15}$$

$$\text{s.t.} \quad \sum_{e=1}^{n} f_e = 1 \tag{16}$$

Applying the method of Lagrange multipliers to the Lagrangian function

$$\mathcal{L}(f, \mu) = \sum_{e=1}^{n} p_e \log \left( \tilde{\beta}_e f_e + \epsilon(1 - f_e) \right) + \mu \left( 1 - \sum_{e=1}^{n} f_e \right), \tag{17}$$

we obtain the following set of equations:

$$0 = \frac{\partial \mathcal{L}}{\partial f_e} = \frac{p_e(\tilde{\beta}_e - \epsilon)}{\tilde{\beta}_e f_e + \epsilon(1 - f_e)} - \mu, \qquad e = 1 \ldots n, \tag{18}$$

$$0 = \frac{\partial \mathcal{L}}{\partial \mu} = 1 - \sum_{e=1}^{n} f_e. \tag{19}$$

Linearizing (*Equation 18*) yields

$$\mu = \frac{p_e}{f_e}\left(1 - \frac{\epsilon}{\tilde{\beta}f_e} + O(\epsilon^2)\right), \tag{20}$$

which we may solve to obtain

$$f_e = \frac{p_e \pm p_e}{2\mu} \mp \frac{\epsilon}{\tilde{\beta}_e} + O(\epsilon^2). \tag{21}$$

We reject the solution $f_e = \epsilon/\tilde{\beta}_e$ on grounds of non-normalizability. Plugging into (19) then yields

$$1 = \sum_{e=1}^{n} f_e = \frac{1}{\mu} - \epsilon \underbrace{\sum_e \frac{1}{\tilde{\beta}_e}}_{z} \implies \mu = \frac{1}{1 + \epsilon z}, \tag{22}$$

from which it follows that

$$f_e = p_e(1 + \epsilon z) - \frac{\epsilon}{\tilde{\beta}_e} + O(\epsilon^2), \tag{23}$$

as desired.

As expected, this asymptotic solution reduces to that of Proposition 3 in the limit of $\epsilon \to 0$. However, at finite $\epsilon$, we find that the optimal probability allocated to phenotype $e$ is now reduced by an amount inversely proportional to that phenotype's on-diagonal fitness $\tilde{\beta}_e$. Thus, unlike in the fully diagonal case, the optimal phenotype distribution ($f_e$) under an assumption of nonzero baseline fitness depends on the precise fitness landscape. In particular, relatively low-fitness phenotypes should now occur *less* frequently than their matching environments. The corresponding Kelly-optimal growth rate is

$$\lambda \simeq \tilde{\lambda} + \epsilon\left(z - \sum_{e=1}^{n} \frac{p_e}{\tilde{\beta}_e}\right) \tag{24}$$

$$= \tilde{\lambda} + \epsilon \sum_e \frac{1 - p_e}{\tilde{\beta}_e}, \tag{25}$$

where $\tilde{\lambda}$ is the optimal growth rate (*Equation 10*) in the case of purely diagonal fitness.

## A continuum of states

We consider the case in which the state space $E$ is uncountably infinite, indexed by a single scalar $e$, so that the environment ($e_t$) becomes a real-valued (though still discrete-time and stationary) stochastic process. We assume that the phenotype space is continuous and isomorphic to $E$, such that it, too, can be indexed by $e$. Then the categorical distributions ($p_e$) and ($f_\phi$) of environments and phenotypes, respectively, become density functions $p(e)$ and $f(\phi)$. Likewise, the fitness matrix $\beta_{e\phi}$ becomes a continuous function $\beta(e, \phi)$ whose bandwidth controls the 'sharpness' or 'fuzziness' of the correspondence between environmental states and phenotypes. By analogy to *Equation 1*, we have

$$G_k = \prod_{t=1}^{k} \int_E \beta(e_t, \phi)f(\phi)\mathrm{d}\phi, \tag{26}$$

where $e_t$ are i.i.d. samples from the distribution with density $p(e)$. One possible form of $\beta(e, \phi)$ is proportional to a Gaussian with standard deviation $\sigma$ centered at $\phi = e$:

$$\beta(e, \phi) = \frac{\hat{\beta}(e, \phi)}{\sqrt{2\pi\sigma^2}} e^{-\frac{(\phi - e)^2}{2\sigma^2}}. \tag{27}$$

When $\sigma$ is small, we may use Laplace's method (equivalent to considering the limit $\sigma \to 0$) to write

$$G_k = \prod_{t=1}^{k} \int_E \beta(e_t, \phi) f(\phi) \mathrm{d}\phi \tag{28}$$

$$= \prod_{t=1}^{k} \int_E \frac{\hat{\beta}(e_t, \phi) f(\phi)}{\sqrt{2\pi\sigma^2}} e^{-\frac{(\phi - e_t)^2}{2\sigma^2}} \mathrm{d}\phi \tag{29}$$

$$= \prod_{t=1}^{k} \left[ \hat{\beta}(e_t, e_t) f(e_t) + O(\sigma^2) \right]. \tag{30}$$

We have thus reduced the problem to the discrete case of (2) treated above, so that the analogue of Proposition 3 follows trivially.

## Claim 2

For $\sigma$ small, the phenotype distribution $f(\phi)$ that maximizes the asymptotic growth rate $\lambda$ is just $f(\phi) \simeq p(e)$, independent of the fitness landscape $\beta(e, \phi)$.

