## [Editor Report · eLife Assessment]

Maloney et al. offer an **important** contribution to understanding the potential ecological mechanisms behind individual behavioral variation. By providing **compelling** theoretical and experimental data, the study bridges the gap between individual, apparently stochastic behavior with its evolutionary purpose and consequences. The work further provides a testable and generalizable model framework to explore behavioral drift in other behaviors.

---

## [Referee Report · Reviewer #1 (Public review)]

Summary:

In "Drift in Individual Behavioral Phenotype as a Strategy for Unpredictable Worlds," Maloney et al. (2026) investigate changes in individual responses over time, referred to as behavioral drift within the lifespan of an animal. Drift, as defined in the paper, complements stable behavioral variation (animal individuality/personality within a lifetime) over shorter timeframes, which the authors associate with an underlying bet-hedging strategy. The third timeframe of behavioral variability that the authors discuss occurs within seasons (across several generations of some insects), termed "adaptive tracking." This division of "adaptive" behavioral variability over different timeframes is intuitively logical and adds valuable depth to the theoretical framework concerning the ecological role of individual behavioral differences in animals.

Strengths:

While the theoretical foundations of the study are compelling, the connection between the experimental data (Fig. 1) and the modeling work (Fig. 2-4) is convincing.

Weaknesses:

In the experimental data (Fig. 1), the authors describe the changes in behavioral preferences over time. While generally plausible, I had identified three significant issues with the experiments that were addressed in the revision:

(1) All of the subsequent theoretical/simulation data is based on changing environments, yet all the experiments are conducted in unchanging environments. While this may suffice to demonstrate the phenomenon of behavioral instability (drift) over time, it does not fully link to the theory-driven work in changing environments. A full experimental investigation of this would be beyond the scope of the current work.

(2) The temporal aspect of behavioral instability has been addressed in Figure 1F.

(3) The temporal dimension leads directly into the third issue: distinguishing between drift and learning (e.g., line 56). This issue has been further discussed in the revised manuscript.

---

## [Referee Report · Reviewer #2 (Public review)]

Summary:

This is an inspired study that merges the concept of individuality with evolutionary processes to uncover a new strategy that diversifies individual behavior that is also potentially evolutionarily adaptive.

The authors use time-resolved measurement of spontaneous, innate behavior, namely handedness or turn bias in individual, isogenic flies, across several genetic backgrounds.

They find that an individual's behavior changes over time, or drifts. This has been observed before, but what is interesting here is that by looking at multiple genotypes, the authors find the amount of drift is consistent within genotype i.e., genetically regulated, and thus not entirely stochastic. This is not in line with what is known about innate, spontaneous behaviors. Normally, fluctuations in behavior would be ascribed to a response to environmental noise. However, here, the authors go on to find what is the pattern or rule that determines the rate of change of the behavior over time within individuals. Using modeling of behavior and environment in the context of evolutionarily important timeframes such as lifespan or reproductive age, they could show when drift is favored over bet-hedging and that there is an evolutionary purpose to behavioral drift. Namely, drift diversifies behaviors across individuals of the same genotype within the timescale of lifespan, so that the genotype's chance for expressing beneficial behavior is optimally matched with potential variation of environment experienced prior to reproduction. This ultimately increases fitness of the genotype. Because they find that behavioral drift is genetically variable, they argue it can also evolve.

Strengths:

Unlike most studies of individuality, in this study, authors consider the impact of individuality on evolution. This is enabled by the use of multiple natural genetic backgrounds and an appropriately large number of individuals to come to the conclusions presented in the study. I thought it was really creative to study how individual behavior evolves over multiple timescales. And indeed this approach yielded interesting and important insight into individuality. Unlike most studies so far, this one highlights that behavioral individuality is not a static property of an individual, but it dynamically changes. Also, placing these findings in the evolutionary context was beneficial. The conclusion that individual drift and bet-hedging are differently favored over different timescales is, I think, a significant and exciting finding.

Overall, I think this study highlights how little we know about the fundamental, general concepts behind individuality and why behavioral individuality is an important trait. They also show that with simple but elegant behavioral experiments and appropriate modeling, we could uncover fundamental rules underlying the emergence of individual behavior. These rules may not at all be apparent using classical approaches to studying individuality, using individual variation within a single genotype or within a single timeframe.

Weaknesses:

I am unconvinced by the claim that serotonin neuron circuits are regulating behavioral drift, especially because of its bidirectional effect and lack of relative results for other neuromodulators. Without testing other neuromodulators, it will remain unclear if serotonin intervention increases behavioral noise within individuals, or if any other pharmacological or genetic intervention would do the same. Another issue is that the amount of drugs that the individuals ingested was not tracked. Variable amounts can result in variable changes in behavior that are more consistent with the interpretation of environmental plasticity, rather than behavioral drift. With the current evidence presented, individual behavior may change upon serotonin perturbation, but this does not necessarily mean that it changes or regulates drift.

However, I think for the scope of this study, finding out whether serotonin regulates drift or not is less important. I understand that today there is a strong push to find molecular and circuit mechanisms of any behavior, and other peers may have asked for such experiments, perhaps even simply out of habit. Fortunately, the main conclusions derived from behavioral data across multiple genetic backgrounds and the modeling are anyway novel, interesting and in fact more fundamental than showing if it is serotonin that does it or not.

To this point, one thing that was unclear from the methods section is whether genotypes that were tested were raised in replicate vials and how was replication accounted for in the analyses. This is a crucial point - the conclusion that genotypes have different amounts of behavioral drift cannot be drawn without showing that the difference in behavioral drift does not stem from differences in developmental environment.

Comments on the latest version:

The changes to the manuscript sufficiently addressed my few comments. I do not have anything else substantial to add to my review and I am comfortable with my initial assessment.

---

## [Referee Report · Reviewer #3 (Public review)]

The paper begins by analyzing the drift in individual behavior over time. Specifically, it quantifies the circling direction of freely walking flies in an arena. The main takeaway from this dataset is that while flies exhibit an individual turning bias (when averaged over time), yet their preferences fluctuate over slow timescales.

To understand whether genetic or neuromodulatory mechanisms influence the drift in individual preference, the authors test different fly strains in a Y maze concluding that both genetic background and the neuromodulator serotonin contribute to the degree of drift (although with some contrasting results). The use of a different assay for this different dataset (Y maze istead of wide arena) is justified by previous observation of similar behavioral biases in these assay. Yet the conceptual link between the spectral power analysis used for the first dataset and the autoregressive model used for the second remains unclear.

Finally, the authors use theoretical approaches to show the potential advantage of individual drift for survival in unpredictable, fluctuating environments. They demonstrate that while bet-hedging provides an advantage over timescales matching the generation time (since reproduction is required), it offers less benefit on shorter timescales, where an increased individual drift could be advantageous.

---

## [Author Response]

The following is the authors’ response to the original reviews.

**Public Reviews:**

**Reviewer #1 (Public review):**
Summary:In "Drift in Individual Behavioral Phenotype as a Strategy for Unpredictable Worlds," Maloney et al. (2024) investigate changes in individual responses over time, referred to as behavioral drift within the lifespan of an animal. Drift, as defined in the paper, complements stable behavioral variation (animal individuality/personality within a lifetime) over shorter timeframes, which the authors associate with an underlying bet-hedging strategy. The third timeframe of behavioral variability that the authors discuss occurs within seasons (across several generations of some insects), termed "adaptive tracking." This division of "adaptive" behavioral variability over different timeframes is intuitively logical and adds valuable depth to the theoretical framework concerning the ecological role of individual behavioral differences in animals.Strengths:While the theoretical foundations of the study are strong, the connection between the experimental data (Figure 1) and the modeling work (Figure 2-4) is less convincing.Weaknesses:In the experimental data (Figure 1), the authors describe the changes in behavioral preferences over time. While generally plausible, I identify three significant issues with the experiments:(1) All of the subsequent theoretical/simulation data is based on changing environments, yet all the experiments are conducted in unchanging environments. While this may suffice to demonstrate the phenomenon of behavioral instability (drift) over time, it does not properly link to the theory-driven work in changing environments. An experiment conducted in a changing environment and its effects on behavioral drift would improve the manuscript's internal consistency and clarify some points related to (3) below.

We have added further discussion of this to the discussion section.

(2) The temporal aspect of behavioral instability. While the analysis demonstrates behavioral instability, the temporal dynamics remain unclear. It would be helpful for the authors to clarify (based on graphs and text) whether the behavioral changes occur randomly over time or follow a pattern (e.g., initially more right turns, then more left turns). A proper temporal analysis and clearer explanations are currently missing from the manuscript.

We have added a figure (1F to better visualize the changes in handedness over days). We have also pointed out the connection between the power spectrum and the autoregressive model given by the Wiener-Khinchen theorem (which states that the autocorrelation function of a wide-sense stationary process has a spectral decomposition of its power spectrum).

(3) The temporal dimension leads directly into the third issue: distinguishing between drift and learning (e.g., line 56). In the neutral stimuli used in the experimental data, changes should either occur randomly (drift) or purposefully, as in a neutral environment, previous strategies do not yield a favorable outcome. For instance, the animal might initially employ strategy A, but if no improvement in the food situation occurs, it later adopts strategy B (learning). In changing environments, this distinction between drift and learning should be even more pronounced (e.g., if bananas are available, I prefer bananas; once they are gone, I either change my preference or face negative consequences). Alternatively, is my random choice of grapes the substrate for the learning process towards grapes in a changing environment? Further clarification is needed to resolve these potential conflicts.

We have discussed this further in the discussion.

**Reviewer #2 (Public review):**
Summary:This is an inspired study that merges the concept of individuality with evolutionary processes to uncover a new strategy that diversifies individual behavior that is also potentially evolutionarily adaptive.The authors use a time-resolved measurement of spontaneous, innate behavior, namely handedness or turn bias in individual, isogenic flies, across several genetic backgrounds.They find that an individual's behavior changes over time, or drifts. This has been observed before, but what is interesting here is that by looking at multiple genotypes, the authors find the amount of drift is consistent within genotype i.e., genetically regulated, and thus not entirely stochastic. This is not in line with what is known about innate, spontaneous behaviors. Normally, fluctuations in behavior would be ascribed to a response to environmental noise. However, here, the authors go on to find what is the pattern or rule that determines the rate of change of the behavior over time within individuals. Using modeling of behavior and environment in the context of evolutionarily important timeframes such as lifespan or reproductive age, they could show when drift is favored over bet-hedging and that there is an evolutionary purpose to behavioral drift. Namely, drift diversifies behaviors across individuals of the same genotype within the timescale of lifespan, so that the genotype's chance for expressing beneficial behavior is optimally matched with potential variation of environment experienced prior to reproduction. This ultimately increases the fitness of the genotype. Because they find that behavioral drift is genetically variable, they argue it can also evolve.Strengths:Unlike most studies of individuality, in this study, the authors consider the impact of individuality on evolution. This is enabled by the use of multiple natural genetic backgrounds and an appropriately large number of individuals to come to the conclusions presented in the study. I thought it was really creative to study how individual behavior evolves over multiple timescales. And indeed this approach yielded interesting and important insight into individuality. Unlike most studies so far, this one highlights that behavioral individuality is not a static property of an individual, but it dynamically changes. Also, placing these findings in the evolutionary context was beneficial. The conclusion that individual drift and bet-hedging are differently favored over different timescales is, I think, a significant and exciting finding.Overall, I think this study highlights how little we know about the fundamental, general concepts behind individuality and why behavioral individuality is an important trait. They also show that with simple but elegant behavioral experiments and appropriate modeling, we could uncover fundamental rules underlying the emergence of individual behavior. These rules may not at all be apparent using classical approaches to studying individuality, using individual variation within a single genotype or within a single timeframe.Weaknesses:I am unconvinced by the claim that serotonin neuron circuits regulate behavioral drift, especially because of its bidirectional effect and lack of relative results for other neuromodulators. Without testing other neuromodulators, it will remain unclear if serotonin intervention increases behavioral noise within individuals, or if any other pharmacological or genetic intervention would do the same. Another issue is that the amount of drugs that the individuals ingested was not tracked. Variable amounts can result in variable changes in behavior that are more consistent with the interpretation of environmental plasticity, rather than behavioral drift. With the current evidence presented, individual behavior may change upon serotonin perturbation, but this does not necessarily mean that it changes or regulates drift.However, I think for the scope of this study, finding out whether serotonin regulates drift or not is less important. I understand that today there is a strong push to find molecular and circuit mechanisms of any behavior, and other peers may have asked for such experiments, perhaps even simply out of habit. Fortunately, the main conclusions derived from behavioral data across multiple genetic backgrounds and the modeling are anyway novel, interesting, and in fact more fundamental than showing if it is serotonin that does it or not.

We have adjusted our wording and contextualized our claims based on previous literature.

To this point, one thing that was unclear from the methods section is whether genotypes that were tested were raised in replicate vials and how was replication accounted for in the analyses. This is a crucial point - the conclusion that genotypes have different amounts of behavioral drift cannot be drawn without showing that the difference in behavioral drift does not stem from differences in developmental environment.

We have reanalyzed the behavioral data in a hierarchical model to account for batch effects. Accounting for batch effects (Fig 1G, S1G) we still observe differences between genotypes and for pharmaceutical manipulations of serotonin, though our data provides more equivocal evidence for the effects of *trh*^n^ on drift.

**Reviewer #3 (Public review):**
Summary:The paper begins by analyzing the drift in individual behavior over time. Specifically, it quantifies the circling direction of freely walking flies in an arena. The main takeaway from this dataset is that while flies exhibit an individual turning bias (when averaged over time), their preferences fluctuate over slow timescales.To understand whether genetic or neuromodulatory mechanisms influence the drift in individual preference, the authors test different fly strains concluding that both genetic background and the neuromodulator serotonin contribute to the degree of drift.Finally, the authors use theoretical approaches to identify the range of environmental conditions under which drift in individual bias supports population growth.Strengths:The model provides a clear prediction of the environmental fluctuations under which a drift in bias should be beneficial for population growth.The approach attempts to identify genetic and neurophysiological mechanisms underlying drift in bias.Weaknesses:Different behavioral assays are used and are differently analysed, with little discussion on how these behaviors and analyses compare to each other.

We have added text indicating that these two behavioral responses have previously been shown to be correlated to each other and that the spectral power analysis and autoregressive model are conceptually linked.

Some of the model assumptions should be made more explicit to better understand which aspects of the behaviors are covered.

We have added a table in the supplemental clarifying all of the parameters of modeling for each figure.

**Recommendations for the authors:**
**Reviewing Editor Comments**:Highlights of the Consultation Session of 3 ReviewersIn the consultation session, the reviewers discussed as particularly important the relative contribution of genotype and variable environment. Further analyses of the replicates of the genotypes were suggested to exclude the environment as the source of difference in the extent of drift between genotypes. If the difference in the extent of drift between replicates is greater than the difference in the extent of drift between genotypes, then one cannot really say that there is a genetic control over drift and that it would evolve (which is still an interesting result, but would be less exciting for a follow-up evolution experiment). If replicates differ, testing whether the relative difference in the extent of drift between genotypes is maintained across environments would also be strong evidence that the extent of behavioral drift is a property of a genotype and not a mere result of a fluctuating/variable environment. The authors do present two behavior paradigms that can serve the purpose of comparing the relative extent of drift between genotypes across the two paradigms that they already have. The authors might consider whether experimental data could be brought closer to theory by including an experiment in a variable environment (e.g temp or diet changes etc.).Reviewers also agreed in the consultation session that methods and definitions were somewhat cryptic, and it would be very helpful if they were more detailed. For example, linking the free walking analysis to the Ymaze and then the model1 to the model2 was not straightforward.

We have added text to make more explicit the theoretical connection between the freewalking analysis, the ymaze analysis, and the model. We have added text and a supplemental table to clarify the methods.

**Reviewer #1 (Recommendations for the authors):**
(1) Line 161: The authors state in the supplement that they used DGRP strains, which are inbred and not isogenic. According to the original authors, they possess 99.3% genetic identity. The isoD1 strain has no known crossing scheme, so complete chromosome isogeneity remains questionable, especially after 12 or more years since its creation. The authors should refer to the strains as "near-isogenic" or a similar term.

We have adjusted the language as suggested to be more accurate.

(2) Lines 276, 338: The manuscript contains some unfinished sentences or remnants from the drafting process (e.g., "REFREF"). A thorough editorial review is recommended to eliminate such errors.

We have cleaned up all references and made additional passes to adjust text.

**Reviewer #2 (Recommendations for the authors):**
(1) If the authors want to claim that serotonin is a regulator of drift, they should provide a negative control experiment, using equivalent perturbations of another neuromodulator and non-modulator. Alternatively, they could simply soften the claims revolving around serotonin and its putative direct role in modulating drift.

We have softened the claims as suggested to avoid claiming our results show a specific role for serotonin.

(2) I would suggest always using "behavioral drift" when referring to drift, especially in the context of modeling, because it can be easily confused with genetic drift and cause confusion when reading.

We have adjusted the language throughout the manuscript per this suggestion.

(3) It would be good to see in the methods if the 2-hour assays were always done at the same time of the fly's subjective day and when (e.g. how many hours after lights on).

We have clarified this.

(4) I understand that many experiments use methodology replicated from the group's previous work, but I would recommend elaborating the experimental methods section in the supplementary such that the reader can understand and reproduce the methods without having to sift through and look for them in previous papers.

We have expanded on our discussion of the methodology in the methods section.

**Reviewer #3 (Recommendations for the authors):**
The paper begins by analyzing the drift in individual behavior over time. Specifically, it quantifies the circling direction of freely walking flies in an arena. The main takeaway from this dataset is that flies exhibit an individual turning bias (when averaged over time), yet their preferences fluctuate over slow timescales. However, it's unclear why the authors chose to switch to a different assay to compare strains. In particular, it's ambiguous whether the behavioral measure in one setup is comparable to that in the other; specifically, whether a bias in one setup reflects the same type of bias in the other. The behavior is also sampled differently across setups (though the details are unclear; see comments below) and analyzed using different methods. Consequently, it remains uncertain whether the slow fluctuations observed in the arena setup are also present in the Y maze. It appears that the analysis of the Y maze data only addresses individual behavioral variance or, at most, day-to-day changes, without accounting for longer-term correlations in bias-which I understood to be the primary interest in the arena setup. Some clarification is needed here (see specific comments below).In Figure 2, the authors attempt to show the potential advantage of individual drift for survival in unpredictable, fluctuating environments. They demonstrate that while bet-hedging provides an advantage over timescales matching the generation time (since reproduction is required), it offers less benefit on shorter timescales, where an increased individual drift could be advantageous. This approach is well-conceived, and the findings are convincing, though the model would benefit from further clarification and additional explanation in the text.Here are some more specific comments:PART 1:(1) L 223 one probably cannot see a circadian peak at 24h if the data were filtered at 24h, did they look with another low pass cutoff?

We clarified in the text that the power spectrum analysis was performed on unfiltered data.

(2) L 243 the spread in standard deviation is said to be consistent with drifting bias, however, I do not agree with this. The variation could be stochastic but independent across days, and show no temporal correlation. As done with the circular arena, a drift should be estimated as a temporal correlation in the behavior.

It is consistent insofar as seeing a non-zero standard deviation is a necessary condition for drift. While it does not show that there is any consistency over time, this can be inferred from the autoregressive model (as well as previous work). We have added text to make this clearer.

(3) In the autoregressive model this temporal aspect seems to be incorporated only to the first order (from day to day). Therefore, from what I understand, the drift term is not correlated over time. This seems very different from the spectral analysis done in the circular assay, and I wonder if it fits at all the initial definition of drift. For example, is the model compatible with a fixed mean and a similar power spectrum as in Figure 1C? The text should clarify that.

can be made clear in the case of σ = 0 and ϕ = 1, where values wouldϕ ≠ be0 In an AR(1) process, datapoints day to day are correlated as long as . This perfectly correlated with each other across time. The AR(1) model and the PSD of circling can be related via the Wiener-Khinchin theorem. We have added text to make this connection clear.

(4) Did serotonin have no role in turning bias? My understanding of previous work was that serotonin should affect the bet-hedg variance as well - the authors should discuss what is expected or not, especially given that the pharmacological and genetic approaches do not have the same effect on bet-edging (Figure 1H-I).

As the pharmacological methods were only applied after eclosion, we do not find it surprising that we do not measure differences in the initially measured distribution of handedness in that case. We do see more evidence of it in the mutations, though the *trh*^n^ experiments provide a less clear effect after our adjustments to account for batch effects.

(5) Methods: It is unclear how flies were handled across days; e.g. in Y mazes: 2h each day for how many days? In the arena flies were imaged either twice daily for 2h per session, or continuously for 24h (L138) - but which data are used where?

We will make this more clear, but all data in figure 1 was the continuous 24h data

This part of the methods is not well explained and I think it should be described in more detail.(6) How many flies per genotype were tested in fig 1E?

Information was added to the caption to duplicate information in the table.

PART 2:(7) In Figure 2B I do not understand the formulation N(50−ϕ: 50, σ), N(phi-et: et, σ) or in general N(x: m, s): does this mean that the variable x has normal distribution with mean m and variance s? Usually this would be written as N(x|m, s) or N(x; m, s)If so then: N(50−ϕ: 50, σ) = N(ϕ: 0, σ) which has mean=0 while the figure caption says "from a normal distribution centred on the long term environmental mean" - what is the long term environmental mean?If this is correct, and, therefore, we are just centering the mean, what about N(et-phi: et, σ)?

Et is the environment at the time, not the mean of the environment (which is 50). We have added more detail in supplementary methods to address this.

(8) Should ϕ vary between 1-100? And is the environmental parameter in Figure 2C also varying between 1-100? These ranges should be written somewhere.

While implied in the sigma notation, we have added more detail in supplementary methods to explain the situation.

(9) As far as I understand the bounding envelope in Figure 2B is necessary to contain the drift model. In Figure 1F, a bounding effect was generated by the "tendency to revert to no bias." It is unclear to me whether these two formulations are equivalent. Moreover, none of these two models might be able to recapitulate the correlations observed in the circular arena and analyzed spectrally in Figure 1C. It would be necessary that the author make an effort to relate these models/quantifications one to another. My understanding of Figure 1B is that there are slow fluctuations around the mean. Is the bounded drift model in 2B not returning to the same mean? And do these models generate slow fluctuations? Further explanation could help clarify these points.

We have added additional explanation to explain the connection between the power spectrum and the two methods of (phi and bounding envelop) of establishing stationarity.

(10) Expanding on the above: I thought that the definition of individuality is based on some degree of stability over days. However, both models assume drift to occur from day to day (and also the analysis of the DGRP lines assumes so). Some clarification here could help: is the initial bet-edging variation maintained in the population? And is the mean individual bias still a thing or it is just drifting away all the time?

The initial bet-hedging is maintained to some degree, based on the parameter of phi and the bounding envelope. We have added text to make this clearer.

(11) In both Figures 2C and 2E the populations are always shrinking, is that correct? And if so, is it expected? Does the model allow growth in a constant environment?

As the plotted values are the log, the optimal environments do allow growth (visible more clearly in 2D). We have added some text to make this clearer.

(12) Growth is quantified only across 100 days (Figure 2D) but at day 100 there is not something like a steady state, how is 100 chosen? Would it make sense to check longer times to see if the system eventually takes off? And if not, why?(13) Related to the above: what is the growth range achieved in Figure 3A-B? Is the heatmap normalized to the same value across conditions? I think it would be important to consider the absolute range of variation of growth or at least the upper value across conditions.Moreover: is growth quantified at day 100? What happens at longer times? Does the temporal profile of the growth curve differ across environmental conditions? (I'm referring to a Figure as 2D).

As we are plotting the log change, we are ultimately showing the growth rate. While a more realistic model would involve carrying capacity, we believe a simplified model showing growth or no growth captures the difference in growth rate between different strategies. We have added some text to make this clearer.

(14) Suddenly at line 502, sexual maturity is introduced as a parameter, which was never mentioned before, called a_min in the figure legend of panel 3a, but it is unclear where this is in the model. And please also clarify if sex maturity is the same as generation time.

Sexual maturity is the same as generation time, we have standardized terminology throughout the paper.

(15) Regarding lines 505-508, could one simply conclude that in this model formulation, the generation time has the effect of a low pass filter on environmental fluctuation? The question is: is this filtering effect the only effect of generation time?

While this seems to capture the high-frequency effect we see, it does not explain the shift from bet-hedging->drift we see at lower-frequency environmental fluctuations.

(16) What reproductive rate is used for the PCA analysis? Is the variance associated with the drift so low because of choosing a fast reproductive rate? A comment in the main text would be helpful.

We have clarified that these plots were done at 10 days.